# Inhibitors of ApiAP2 protein DNA binding exhibit multistage activity against *Plasmodium* parasites

**Timothy James Russell**[1,2,3,4], **Erandi K. De Silva**[5], **Valerie M. Crowley**[1], **Kathryn Shaw-Saliba**[6], **Namita Dube**[1], **Gabrielle Josling**[1,3,4], **Charisse Flerida A. Pasaje**[7], **Irene Kouskoumvekaki**[8], **Gianni Panagiotou**[9,10], **Jacquin C. Niles**[7], **Marcelo Jacobs-Lorena**[6], **C. Denise Okafor**[1,11], **Francisco-Javier Gamo**[12], **Manuel Llinás**[1,2,3,4,11] *

1 Department of Biochemistry and Molecular Biology, Pennsylvania State University, State College, Pennsylvania, United States of America, 2 Huck Institutes Center for Eukaryotic Gene Regulation (CEGR), Pennsylvania State University, State College, Pennsylvania, United States of America, 3 Huck Institutes Center for Malaria Research (CMaR), Pennsylvania State University, State College, Pennsylvania, United States of America, 4 Huck Institutes Center for Infectious Disease Dynamics, Pennsylvania State University, State College, Pennsylvania, United States of America, 5 Lewis-Singler Institute for Integrative Genomics, Princeton University, Princeton, New Jersey, United States of America, 6 Department of Molecular Biology and Immunology, Malaria Research Institute, Johns Hopkins Bloomberg School of Public Health, Baltimore, Maryland, United States of America, 7 Department of Biological Engineering, Massachusetts Institute of Technology, Cambridge, Massachusetts, United States of America, 8 Department of Systems Biology, Technical University of Denmark, Kongens Lyngby, Denmark, 9 Systems Biology and Bioinformatics, Leibniz Institute for Natural Products Research and Infection Biology, Hans Knöll Institute, Jena, Germany, 10 Department of Medicine, the University of Hong Kong, Hong Kong SAR, China, 11 Department of Chemistry, Pennsylvania State University, State College, Pennsylvania, United States of America, 12 Global Health Medicines, GlaxoSmithKline, Tres Cantos, Spain

* manuel@psu.edu

## Abstract

*Plasmodium* parasites are reliant on the Apicomplexan AP2 (ApiAP2) transcription factor family to regulate gene expression programs. AP2 DNA binding domains have no homologs in the human or mosquito host genomes, making them potential antimalarial drug targets. Using an *in-silico* screen to dock thousands of small molecules into the crystal structure of the AP2-EXP (Pf3D7_1466400) AP2 domain (PDB:3IGM), we identified putative AP2-EXP interacting compounds. Four compounds were found to block DNA binding by AP2-EXP and at least one additional ApiAP2 protein. Our top ApiAP2 competitor compound perturbs the transcriptome of *P. falciparum* trophozoites and results in a decrease in abundance of $\log_2$ fold change > 2 for 50% (46/93) of AP2-EXP target genes. Additionally, two ApiAP2 competitor compounds have multi-stage anti-*Plasmodium* activity against blood and mosquito stage parasites. In summary, we describe a novel set of antimalarial compounds that interact with AP2 DNA binding domains. These compounds may be used for future chemical genetic interrogation of ApiAP2 proteins or serve as starting points for a new class of antimalarial therapeutics.

**Data Availability Statement:** The numerical data used in all figures is included in S1 Data. All remaining data is included in the manuscript and its supporting information files. Raw and

processed sequencing data for AP2-EXP ChIP-seq and whole genome sequencing for transgenic parasite lines AP2-EXP::GFP and AP2-EXP::HA have been deposited in the SRA under PRJNA818769: www.ncbi.nlm.nih.gov/bioproject/?term=PRJNA818769. Raw microarray data for the Compound C and DMSO control RNA time course is available in NCBI GEO GSE202876: https://www.ncbi.nlm.nih.gov/geo/query/acc.cgi?acc=GSE202876.

**Funding:** This work was funded through NIH/NIAID R01AI076276 (M.L.), R01AI125565 (M.L.), and with support from the Center for Quantitative Biology (P50 GM071508) (M.L.). T.J.R. was supported by NIH T32 Predoctoral Training Grant (5T32GM125592-01) awarded to the Center for Eukaryotic Gene Regulation (CEGR) at The Pennsylvania State University. G.J. is a recipient of the Sir Keith Murdoch Fellowship from the American Australian Association and a Postdoctoral Research Grant from the American Heart Association (16POST26420067). The funders had no role in study design, data collection and analysis, decision to publish, or preparation of the manuscript. NIH/NIAID: https://www.niaid.nih.gov/ Center for Quantitative Biology: https://lsi.princeton.edu/research/center-quantitative-biology NIH: https://www.nih.gov/ American Australian Association: https://www.americanaustralian.org/ AHA: https://www.heart.org/.

**Competing interests:** I have read the journal's policy and the authors of this manuscript have the following competing interests: F.J.G. is a GlaxoSmithKline employee and own shares of the company.

## Author summary

*Plasmodium* parasites are the causative agent of malaria, which resulted in over 600,000 deaths in 2021. Due to resistance arising for every antimalarial therapeutic deployed to date, new drug targets and druggable pathways must be explored. To address this concern, we used a molecular docking screen to predict competitors of DNA binding by the parasite specific family of Apicomplexan AP2 (ApiAP2) transcription factor proteins for testing *in vitro* and *in vivo*. We find that ApiAP2 competing compounds have antimalarial activity consistent with the disruption of gene regulation. This work will further our understanding of both the biological role and targetability of parasite transcriptional regulation.

## Introduction

Malaria is a disease caused by intracellular parasites from the genus *Plasmodium* that represents a significant health and economic burden worldwide [1]. The most virulent of the human infectious malaria parasites is *Plasmodium falciparum*, which caused more than 200 million cases of malaria and resulted in over 600 thousand deaths in 2021 [1]. Resistance has been reported for every antimalarial therapeutic deployed to date, necessitating the need for malaria drugs that target new parasite processes [2]. To successfully proliferate, *P. falciparum* parasites must develop through a complex lifecycle that includes intracellular and extracellular stages in both the human and *Anopheles* mosquito hosts [3]. The clinical symptoms of malaria are caused by the intraerythrocytic development cycle (IDC), a 48-hour cyclic asexual proliferation that results in the destruction of red blood cells (RBCs). During the asexual IDC, malaria parasites progress through three morphological stages referred to as ring, trophozoite, and schizont. Parasites transmit from human to mosquito following differentiation and maturation into sexual stage gametocytes. Once ingested by the mosquito, gametes sexually reproduce and ultimately develop into sporozoites, which can be transmitted back to the human host to initiate a liver stage infection which precedes the asexual blood stages [3].

Up to 80% of *P. falciparum* protein coding transcripts are developmentally regulated during the IDC [4–8] as part of a 'just in time' cascade [5,8,9]. This is predicted to be principally driven by the 27 Apicomplexan Apetala AP2 (ApiAP2) proteins [10–12], which are the major family of sequence specific transcription factors encoded in the *Plasmodium* genome. ApiAP2 proteins contain one to three AP2 DNA binding domains that are analogous to the plant APE-TALA2/Ethylene Responsive Factor (AP2/ERF) domain and therefore have no homologs in the human or mosquito genome [13,14]. Over half of the ApiAP2 proteins are predicted to be essential for the IDC in *P. falciparum* [15–17]. In-depth studies of ApiAP2 proteins to date have demonstrated essential roles in transcriptional activation or repression throughout parasite development affecting diverse processes including invasion [18,19], heterochromatin maintenance [17,20–23], sexual and mosquito stage differentiation [16,24–34] and heat stress tolerance [35]. Despite these properties which make ApiAP2 proteins potential drug targets, no efforts to date have focused on targeting the sequence specific DNA binding ApiAP2 transcription factors.

Although specifically targeting transcription factors *via* small molecules is challenging, this approach is the subject of advances in the anticancer therapeutic field [36]. Major approaches described to date include inhibition of the protein-ligand interaction interface of nuclear hormone receptors [37] and selective competition of the transcription factor's cognate DNA sequence [38]. Direct inhibition of DNA binding *via* the protein-DNA interaction interface is

a less commonly employed strategy due to the difficulty of designing ligands for a DNA binding interface compared to ligand or protein interaction domains [36]. However, a small molecule based disruption of DNA binding was recently described for the oncogenic transcription factor STAT3, thereby demonstrating the feasibility of this approach [39].

In this study we conducted an *in silico* chemical screen against the crystal structure of the AP2-EXP AP2 domain [40] to predict competitors of DNA binding activity. AP2-EXP is predicted to be essential for the *P. falciparum* asexual blood stage [16,17,21], and its orthologue in the rodent malaria parasite *P. berghei* (PbAP2-Sp) is a master regulator of sporogony [30]. Several compounds selected in the screen can prevent DNA binding by the AP2-EXP AP2 domain, as well as additional AP2 domains, *in vitro*. We find that modifications to the substitution state of an AP2-EXP AP2 domain interfering compound modulates its ability to compete DNA binding *in vitro*. Next, we focused on determining whether AP2 domain proteins are affected *in vivo* upon *Plasmodium* parasite compound exposure. We determined that two ApiAP2 competing compounds cause a similar phenotype when exposed to asexual blood stage *P. falciparum*. To evaluate whether AP2-EXP is among the proteins affected *in vivo*, we measured AP2-EXP genomic occupancy by chromatin immunoprecipitation followed by high throughput sequencing (ChIP-seq) and found high correlation between dysregulated mRNA transcripts in the presence of our lead compound and AP2-EXP gene targets. Finally, two AP2 domain competitor compounds are active against *P. berghei* development in the mosquito stage. Overall, our results demonstrate the potential to chemically target the DNA binding activity of ApiAP2 proteins.

## Results

### *In silico* prediction of AP2-EXP competitors

To identify competitors of DNA binding by an ApiAP2 protein, we used AP2-EXP (PF3D7_1466400), which is the only ApiAP2 protein whose AP2 domain is structurally characterized (PDB accession: 3IGM) [40]. *In silico* molecular docking was run on AUTODOCK [41] using over ten thousand small molecules from both the Tres Cantos Antimalarial Set (TCAMS) [42] and the Drug Bank [43] against AP2-EXP. Docking hits were prioritized based on the free energy of interaction and proximity to amino acids necessary for DNA contact (**S1A Fig**). In some cases, the compound that generated a high docking score was not available for purchase, so alternative choices with >0.9 Tanimoto similarity score were identified, computationally docked, and used for further evaluation (**All compounds listed in S1B Fig**). These docking simulations resulted in a final list of nine top scoring predicted competitors of DNA binding by AP2-EXP, and each compound was assigned an alphabetical identifier A-I (**Figs 1A, S2A, and S2B**) for use in this study. We determined that all nine compounds kill asexual *P. falciparum* parasites at micromolar concentrations (range 11.33–198.90μM) in a growth inhibition assay (**Figs 1A and S3**).

### Four compounds have activity against AP2-EXP *in vitro*

The nine predicted competitors were tested to assess their ability to disrupt AP2-EXP DNA binding using a competitive electrophoretic mobility shift assay (EMSA) (**S4A Fig, DNA oligonucleotides in S1 Table**). Four of the nine compounds (A, B, C and I) (**Fig 1B**) were able to effectively prevent DNA binding by AP2-EXP *in vitro*. Three of the four AP2-EXP competitors have a benzoxazole as the core moiety (Compounds A, B, and C), and Compound I is made up of planar rings. However, Compound D does not affect AP2-EXP DNA binding despite having a benzoxazole group, indicating that this is not the sole determinant of activity (**Fig 1C**). Compounds A, B and C also have the lowest $IC_{50}$ values measured amongst the original candidates

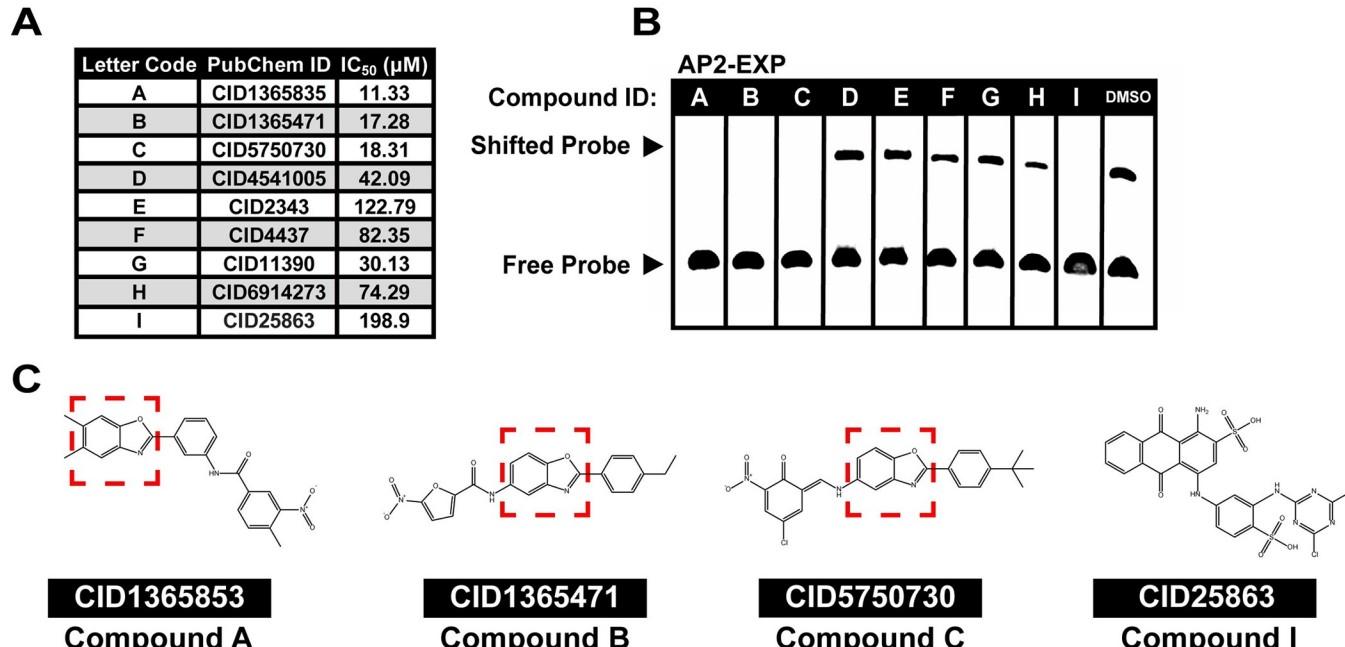

**Fig 1. Nine putative competitors of DNA binding by AP2-EXP were selected by an *in-silico* screen and tested *in vivo* and *in vitro*.** A) Each compound that was prioritized based on the *in-silico* screen was assigned an identifier (A-I) and tested for anti-*Plasmodium* activity in a 48-hour growth inhibition assay against asexual blood stage *P. falciparum*. B) Each putative AP2-EXP competitor was added to an EMSA containing the AP2-EXP AP2 domain. Activity against AP2-EXP will result in a loss of the shifted DNA probe. DMSO vehicle was used as a control for normal DNA binding by AP2-EXP. 150 fmoles of DNA probe, 125ng of AP2-EXP, and 300μM of each compound were used for each lane. C) Chemical structures of Compounds A, B, C, and I. Three of the four compounds that prevent AP2-EXP DNA binding *in vitro* (Compounds A, B, and C) have a benzoxazole core moiety, denoted by a red box.

(11.33–18.31μM), while Compound I has the highest recorded $IC_{50}$ value at 198.90μM (**Fig 1A**).

To assess whether the observed inhibition of ApiAP2 domain DNA binding was specific to the AP2-EXP AP2 domain or might be pan-ApiAP2, we repeated competitive EMSAs with three different purified *P. falciparum* AP2 domains: AP2-I Domain 3 (AP2-I D3) [12,18] (**S4B Fig**), AP2-HS Domain 1 (AP2-HS D1) [12,35] (**S4C Fig**), and PfSIP2 Domain 1 (PfSIP2 D1) [20] (**S4D Fig**) (**S1 Table**). We found that Compounds A, B, C, and I can prevent DNA binding by AP2-I D3 (**S5A Fig**), Compounds B and I prevent DNA binding by AP2-HS D1 (**S5B Fig**) and Compounds B, C and I prevent DNA binding by PfSIP2 D1 (**S5C Fig**).

To assess the relative specificity for ApiAP2 proteins, we performed competitive EMSAs using two non ApiAP2 proteins: the *Arabidopsis thaliana* AP2/ERF domain protein Ethylene Response Factor 1 [44] (AtERF1) (**S4E Fig**) and the human High Mobility Group Box (HMGB) domain protein SOX2 (**S4F Fig**) [45]. Only Compound I was able to compete AtERF1 (**S5D Fig**). SOX2 was competed by Compound A (**S5E Fig**). The lack of AtERF1 inhibition is surprising, given the high structural similarity between AP2-EXP and AtERF1 [44] (**S5F Fig**). SOX2 has no structural similarity to AP2-EXP [46] (**S5F Fig**). Therefore, Compounds B and C have specificity for ApiAP2 protein domains relative to the two protein domains tested *in vitro*. Finally, we performed a competitive EMSA against AP2-EXP using antimalarial drugs with known modes of action: chloroquine, artemisinin, and pyrimethamine [2]. None of these drugs affects AP2-EXP DNA binding, providing evidence that the EMSA assay used can differentiate between putative ApiAP2 competitors and antimalarials with different targets (**S5G Fig**). Pyrimethamine, chloroquine, and artemisinin are diverse in chemical

structure (**S5H Fig**), therefore the assay is not likely to be biased towards a specific class of antimalarial. (EMSA Results Summarized in **S2 Table**).

Each compound identified in the *in-silico* screen contains planar rings which could potentially intercalate DNA [47]. To eliminate the possibility that the four ApiAP2 competing compounds interfere non-specifically with DNA binding proteins by intercalating with DNA, we conducted an ethidium bromide exclusion assay [48]. Compared to the positive control major groove intercalator DRAQ5 [48], only Compound I was able to effectively displace ethidium bromide from DNA (**S6 Fig**). Based on their relative specificity for ApiAP2 proteins compared to Compounds A and I, we focused on Compounds B and C for further study.

## Structural analogs of Compound B differ in activity against AP2-EXP *in vitro*

Our initial competitive EMSA results established that the benzoxazole moiety is overrepresented but not the sole determinant of activity against ApiAP2 proteins (**Fig 1B and 1C**). Therefore, to elucidate additional factors which influence compound activity *in vitro*, we tested the four closest available analogs to Compound B from the TCAMS based on similarity score [42] (Designated as Compounds B-1, B-2, B-3 and B-4) against AP2-EXP. Compounds B-1 and B-4 prevent DNA binding by AP2-EXP (**Fig 2A**) and AP2-I D3 (**S7A Fig**), while Compounds B-2 and B-3 do not prevent DNA binding by either AP2-EXP or AP2-I D3 (**Figs 2A and S7A**). Compound B-1 is the least effective against AP2-EXP DNA binding relative to Compounds B and B-4 (**Fig 2B**). Non-AP2 domain competitor Compounds B-2 and B-3 each have halogen atoms substituted on the benzene ring. Conversely, Compounds B, B-1, and B-4 have methyl or ethyl groups substituted onto the benzene ring and can prevent DNA binding *in vitro* (**Fig 2C**). Compound B-1 has both a methyl group and a chlorine atom substitution. If the halogen atom substitution decreases activity against DNA binding, this mix of substitutions is consistent with its lower efficacy against ApiAP2 proteins *in vitro* relative to Compounds B and B-4. Compounds B, B-1 and B-4 have no difference in DNA intercalation ability measured by ethidium bromide exclusion (**S7B Fig**). A molecular dynamics simulation of AP2-EXP docked with Compound B and each analogue predicts that Compounds B, B-1, B-2, and B-4 dock stably with AP2-EXP, while non-AP2 competitor Compound B-3 moves away from the protein (**S1–S5 Movies**). In summary, substituting bulky, electronegative groups onto Compound B can abolish its activity against AP2-EXP *in vitro*.

## Putative ApiAP2 inhibitor Compounds B and C maximally affect *P. falciparum* trophozoites

To measure the phenotypic defects caused by putative ApiAP2 inhibiting compounds, we determined the parasite killing phenotypes of Compounds B and C against *P. falciparum* during the IDC. Compound H was used as a control since it does not compete AP2-EXP *in vitro* (**Fig 1B**). Each compound was spiked (40μM) into synchronous asexual blood stage *P. falciparum* parasites at 6 hpi and parasitemia was recorded at 6-hour intervals from 6 to 48 hpi (**Fig 3A–3D**). Incubation with Compound H caused an observable developmental defect starting at 42 hpi in which fewer parasites entered schizogony (**Fig 3D and 3E**). In contrast, treatment with either Compound B or C resulted in parasites that similarly failed to progress into mature trophozoites (**Fig 3B, 3C and 3E**), suggesting that Compounds B and C may share a common mode of action.

Due to its commercial availability, we continued with Compound C as the lead compound for phenotypic and molecular characterization. To better determine the timing of the antimalarial action of Compound C, 40μM Compound C was spiked into a highly synchronous

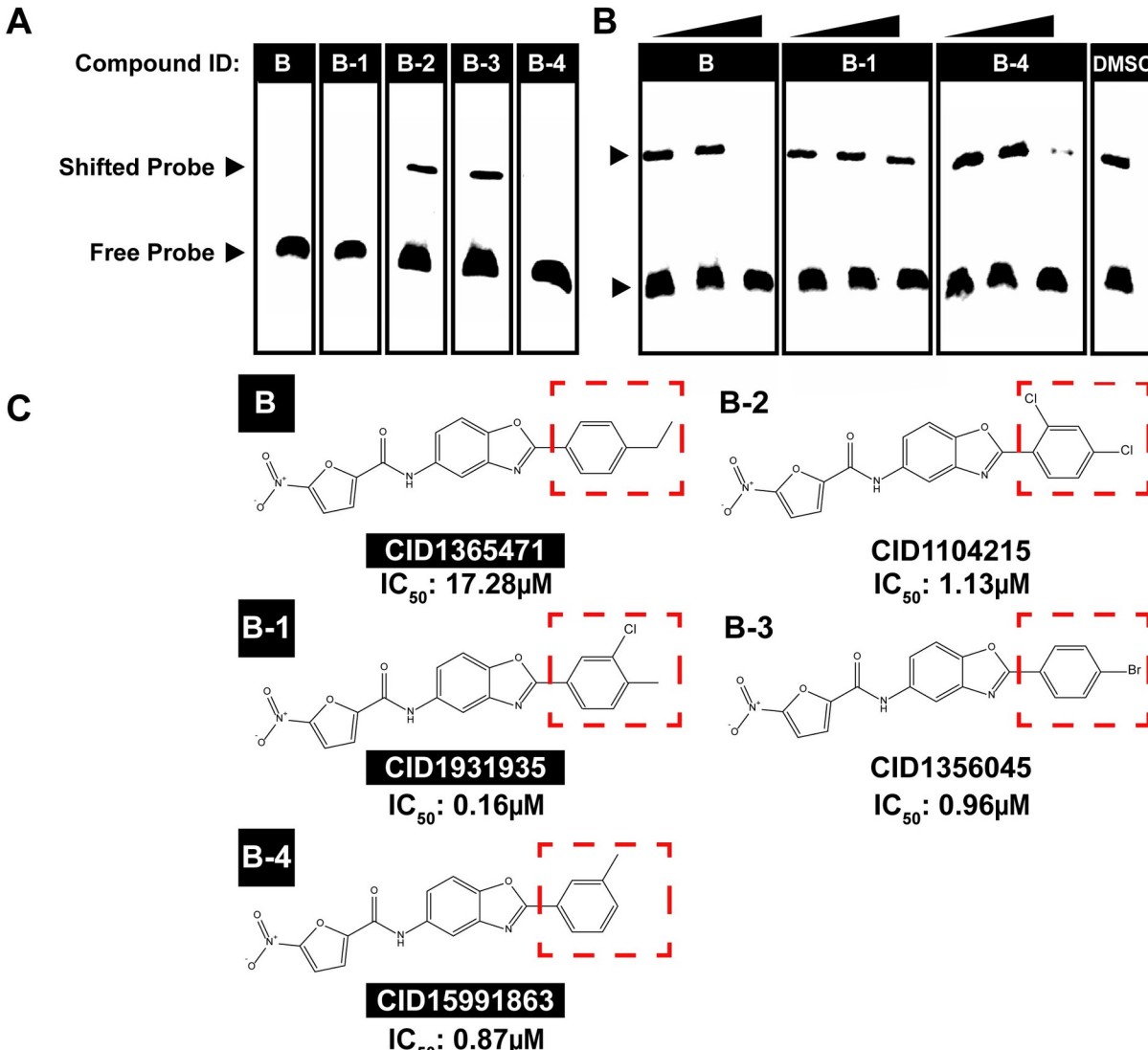

**Fig 2. Analogues of Compound B prevent AP2-EXP DNA binding with differing efficacy.** A) Compounds B, B-1, B-2, B-3, and B-4 were added to an EMSA with AP2-EXP. Activity against AP2-EXP will result in the loss of the shifted probe. 40 fmoles of DNA probe, 125ng of AP2-EXP, and 300μM of each compound were used for each lane. B) Compounds B, B-1 and B-4 were titrated at 25, 50, and 150μM into an EMSA with AP2-EXP to determine whether there are differences in efficacy against AP2-EXP. DMSO vehicle was used as a control for normal DNA binding. 40 fmoles of DNA probe and 125ng of AP2-EXP were used for each lane. The concentration increase for each compound is indicated by the triangle from left to right. C) Chemical structures of Compounds B, B-1, B-2, B-3, and B-4. Compounds that prevent DNA binding by AP2-EXP are highlighted in black. Each compound has a different substitution pattern on the right-side benzene ring, denoted by a red box. The $IC_{50}$ against asexual *P. falciparum* of compounds B-1, B-2, B-3, and B-4 measured by Gamo *et al* [42] is indicated at the bottom of each identifier. The $IC_{50}$ of Compound B was determined in this study.

asexual blood stage *P. falciparum* culture for 8-hour intervals and then washed out starting at 6 hpi (**S8A Fig**). Since the phenotype caused by continuous Compound C exposure manifested as a complete arrest in the early trophozoite stage, resulting in no reinvasion at 48 hpi (**Fig 3C and 3E**), we used the ratio of reinvaded rings at 54 hpi to trophozoites remaining in the culture at 54 hpi as a metric for Compound C induced growth inhibition. With the exception of 38–46 hpi, each interval of Compound C exposure resulted in a significant decrease in the ring to trophozoite ratio (**Fig 3F**). The Compound C spike in interval from 22–30 hpi resulted in the lowest ring to trophozoite ratio amongst the intervals tested (0.54, compared to 9.71 in the DMSO

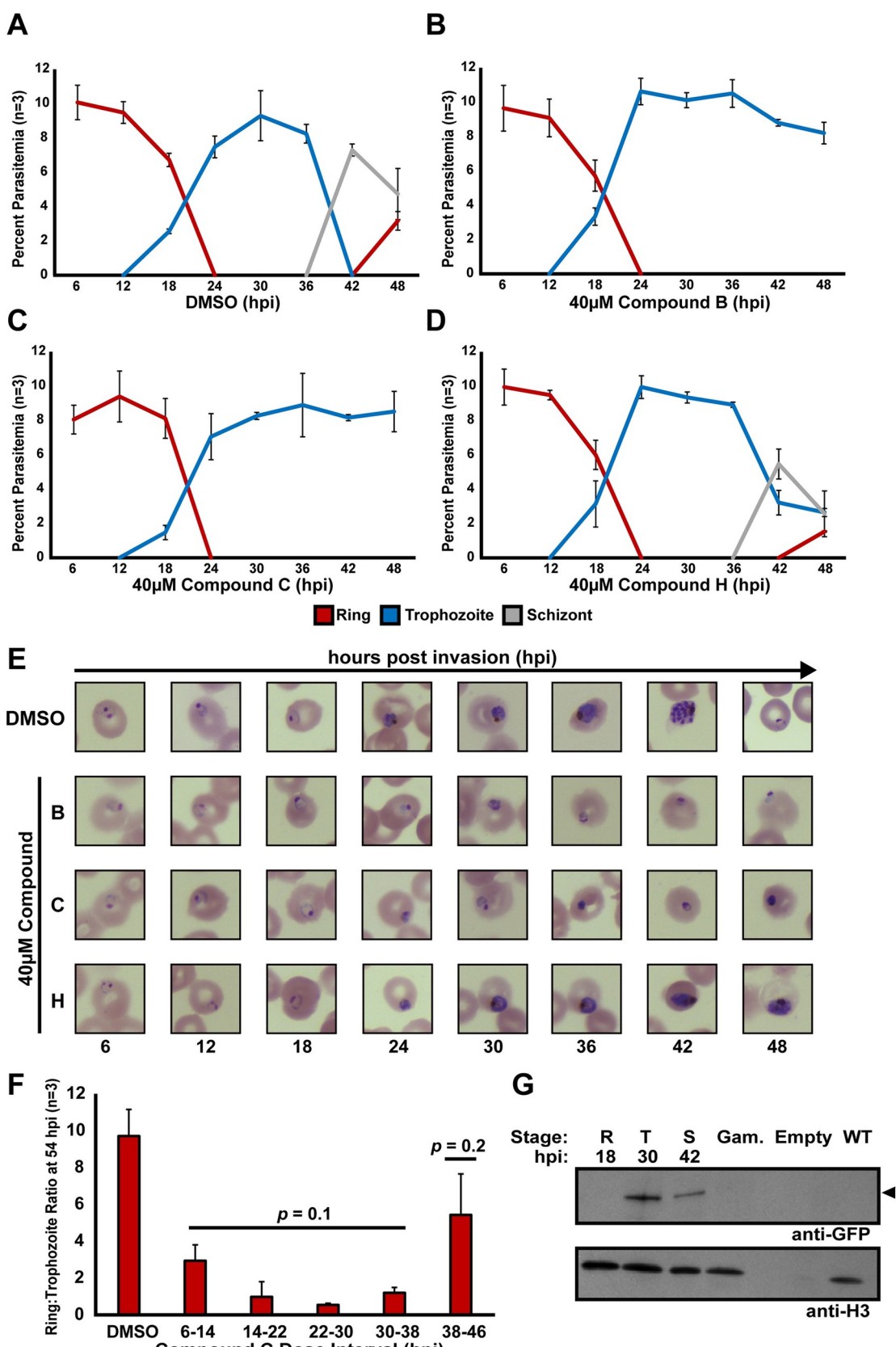

**Fig 3. Compounds B and C affect *P. falciparum* in the mid trophozoite stage, coinciding with the maximum expression of AP2-EXP.** A-D) Highly synchronous wild type Pf3D7 parasites at 10% starting parasitemia were spiked with DMSO vehicle control (A), 40μM Compound B (B), 40μM Compound C (C), or 40μM Compound H (D) and parasite staging was recorded at 6-hour intervals throughout the IDC. Each growth assay was performed in technical and biological triplicate. Error bars represent standard deviation of the mean. The color key indicates the asexual blood stage morphologies recorded at each time point. E) Representative Giemsa-stained parasite images corresponding to the growth assay from panels (A-D). F) Highly synchronous asexual blood stage wild type Pf3D7 parasites at 1% starting parasitemia were dosed with 40μM Compound C for 8-hour intervals starting at 6 hpi. Compound C was removed following each dose interval. At 54 hpi, the ratio of reinvaded rings to stalled trophozoites was recorded. DMSO was used as a vehicle control for normal growth. Each assay was performed in biological and technical triplicate. Error bars represent standard deviation of the mean. Significance of the difference between the DMSO control and Compound C dosed values was tested using the Mann Whitney U-test. G) AP2-EXP protein expression in the asexual blood stages ring (R), trophozoite (T), and schizont (S), and Stage III gametocytes (Gam), was probed by western blot against AP2-EXP endogenously tagged with GFP (AP2-EXP::GFP). The expected molecular weight of AP2-EXP::GFP (147kDa) is indicated by an arrow. Wild type Pf3D7 protein was used as a negative control. The full-length western blot is provided in **S10B Fig**.

control), therefore, the maximum activity of Compound C against asexual blood stage *P. falciparum* occurs between 22–30 hpi (**Figs 3F and S8B**).

## AP2-EXP expression coincides with the growth phenotype of Compound C

In order to determine whether the observed timing of Compound C activity coincides with the presence of AP2-EXP *in vivo*, we tracked its expression throughout the IDC. First, we used Selection Linked Integration (SLI) [49] to create a parasite line with endogenously GFP tagged AP2-EXP (AP2-EXP::GFP) (**S9A Fig**). Using this line, we observed maximum abundance of AP2-EXP::GFP in 30 hpi trophozoites by both western blot and live fluorescence microscopy (**Figs 3G, S10A, and S10B**). AP2-EXP was not detected in maturing Stage III gametocytes (**Fig 3G**). Maximum protein abundance of AP2-EXP in mid-trophozoites was independently confirmed in a parasite line expressing AP2-EXP tagged with 2XHA and the TetR:DOZI mRNA repression aptamer (AP2-EXP::HA) (**Figs S9B and S11**). Therefore, the maximum abundance of AP2-EXP at 30 hpi coincides with the timing of action for Compound C.

## Compound C disrupts the *P. falciparum* transcriptome with stage specificity

Since ApiAP2 proteins are predicted to act as transcription factors [14,17,21] we hypothesized that specific changes in transcript abundance should occur when parasite cultures are exposed to Compound C. We cultured parasites in 12μM ($0.66 \times IC_{50}$) Compound C or DMSO vehicle and measured RNA abundance at seven time points throughout the asexual blood stage. As a quality control, we checked a set of periodically expressed control genes [50,51] (**S12A–S12D Fig**) and the total transcriptome (**S12E–S12F Fig**) of each dataset to monitor the IDC progression. Since 12μM Compound C prevents fewer than 50% of parasites from completing the IDC (**S13A and S13B Fig**), there is not a fully penetrant perturbation to the representation of all IDC stages upon Compound C incubation. The Spearman Correlation between the total transcriptome of DMSO and Compound C samples deviated most significantly at 30 hpi, (Corr. = -0.16) (**Fig 4A**) again coinciding with the observed activity of Compound C (**Fig 3C, 3E, 3F**, Gene Expression values in **S3 Table**).

Next, we quantified differences in transcript abundance that are unique to the progression from 24 to 30 hpi for Compound C vs. DMSO vehicle control populations using Linear Models for Microarray Data (LIMMA [52]) (**Differentially Expressed genes in S3 Table**). Overall, 463 RNA transcripts are decreased in abundance by $\log_2$ fold change >2 between 24–30 hpi in the presence of Compound C. Gene ontology (GO) analysis [53] (**Full GO Terms in S3 Table**) revealed an enrichment for genes that encode proteins important for the parasite to

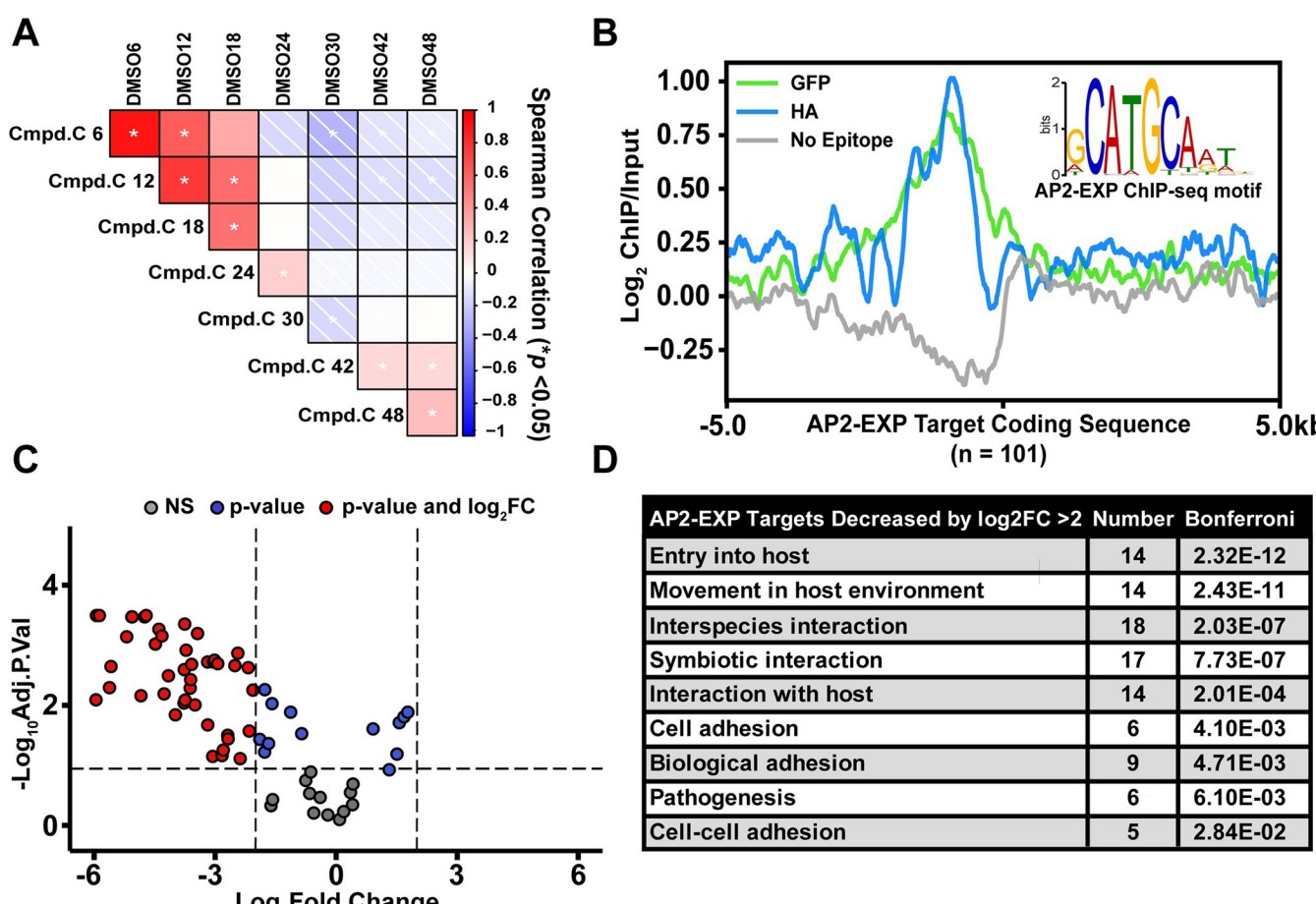

**Fig 4. Compound C disrupts the *P. falciparum* transcriptome specifically at 30 hpi, with bias towards AP2-EXP target genes predicted by ChIP-seq.** A) Highly synchronous wild type Pf3D7 parasites were spiked with 12μM (0.66xIC$_{50}$) Compound C or DMSO vehicle control at 10 hpi. Total RNA was harvested at 6, 12, 18, 24, 30, 42, and 48 hpi for quantification by DNA microarray. The full transcriptome Spearman Correlation between Compound C and DMSO control spiked parasites was plotted as a correlogram. An asterisk indicates *p*-value of correlation < 0.05. B) Three replicates of AP2-EXP Chip-seq were collected at 30 hpi using two genetically tagged parasite lines (AP2-EXP::GFP and AP2-EXP::HA). A no epitope control was collected using wild type Pf3D7 parasites and the anti-GFP antibody. Log$_2$ enrichment of the immunoprecipitate (ChIP) over Input DNA for one replicate of AP2-EXP::GFP, AP2-EXP::HA, and No Epitope Control is plotted relative to the coding sequence start site of AP2-EXP target genes identified in at least 2/3 replicates. CATGCA is the most overrepresented DNA motif within the peaks of AP2-EXP occupancy conserved in at least 2/3 ChIP-seq experiments. C) A volcano plot of the changes in abundance for AP2-EXP target genes at 24–30 hpi in the presence of Compound C. 46/93 detected AP2-EXP target transcripts decrease in abundance by log$_2$ fold change > 2 (Adjusted *p*-value cutoff < 0.1). D) GO-term analysis of transcripts that are both AP2-EXP targets and decreased in abundance with respect to Compound C (Bonferroni *p*-value cutoff < 0.05).

invade or modify red blood cells among decreased transcripts (*e.g.*, RON3, GAP45, RhopH2, RhopH3, MSP1, MSP6). We independently determined differential transcript abundance over the entire time course using the time course specific software Rnits [54] and found high overlap with our LIMMA analysis (**Overlaps in S3 Table**). Therefore, the maximal perturbation of the transcriptome (**Fig 4A**), action of Compound C (**Fig 3C, 3E and 3F**), and maximal abundance of AP2-EXP (**Fig 3G**) all occur at roughly 30 hpi in the IDC.

## AP2-EXP gene targets correlate with differentially abundant transcripts

We used ChIP-seq to determine the genome wide binding occupancy of AP2-EXP. A total of three samples from highly synchronous trophozoites were collected at 30 hpi using both the AP2-EXP::GFP (2 replicates) and AP2-EXP::HA (1 replicate) parasite lines (**Peaks in S4**

Table). As a quality control, we determined that ChIP recovers intact AP2-EXP (S14A Fig), and a co-immunoprecipitation blot confirmed that AP2-EXP interacts with Histone H3 in the nucleus (S14B Fig). In aggregate, AP2-EXP binds 240 genomic loci, corresponding to 101 total target genes (**Genes in S4 Table**). Enrichment of AP2-EXP was well conserved between the GFP and HA tagged parasite lines, as indicated by nearly identical coverage in a metagene plot of AP2-EXP target genes (**Fig 4B**). A 'no epitope control' ChIP done on a wild type Pf3D7 parasite line using the anti-GFP antibody detected only 2 peaks, neither of which overlapped with those of AP2-EXP::GFP or AP2-EXP::HA (**Fig 4B and S4 Table**). AP2-EXP peaks were highly enriched for the known AP2-EXP and PbAP2-Sp DNA motif CATGCA [17,55] (**Figs 4B and S15A–S15C**).

Out of the 463 transcripts decreased in abundance by $\log_2$ fold change > 2 in the presence of Compound C, 46 are AP2-EXP targets (**S3 Table**). This represents 50% (46/93) of AP2-EXP target genes detected in the RNA time course (**Figs 4C and S16**). We found that 7/14 total transcripts decreased in abundance by $\log_2$ fold change > 5, and 16/33 decreased in abundance by $\log_2$ fold change > 4, are AP2-EXP targets (**Fig 4C and S3 Table**). AP2-EXP targets that are dysregulated at 24–30 hpi have functions related to red blood cell invasion and host cell remodeling (**Fig 4D and S3 Table**). To assess whether AP2-EXP DNA binding is impacted by Compound C *in vivo*, parasite cultures were spiked with either 40μM Compound C or DMSO vehicle control at 30 hpi and AP2-EXP occupancy was measured by ChIP-quantitative PCR (ChIP-qPCR). On average, AP2-EXP occupancy was found to decrease at five genomic loci (*gap45*, *sip2*, *ron3*, *ralp*, and *ama1*) in the presence of Compound C compared to non AP2-EXP bound control loci, although these results were not statistically significant (**S17 Fig and S1 Table**).

We then compared AP2-EXP genomic occupancy with several published datasets in order to further evaluate its function as a sequence specific transcription factor. AP2-EXP occupancy is correlated with a nucleosome depleted region (**S18A and S18B Fig**) [56] and activating chromatin marks [57,58] (**S19A and S19B Fig**), and is proximal to target gene transcription start sites (TSS) [59] (**S20 Fig**). The majority of AP2-EXP target genes increase in abundance starting at 32 hpi (**S21 Fig**) [60]. AP2-EXP target gene function is enriched for invasion and red blood cell modification, (**S3 Table**) as was found for the genes that decrease in abundance in the presence of Compound C (**Fig 4D and S3 Table**). While additional targets of Compound C cannot be ruled out, in aggregate, our findings suggest that Compound C inhibits the function of AP2-EXP as a sequence specific transcription factor during *P. falciparum* asexual blood stage development.

## ApiAP2 competitor Compounds B and C are active against mosquito stage *Plasmodium* parasites

Elucidation of the role of essential *P. falciparum* ApiAP2 proteins in the IDC is challenging due to the inability to knock out asexual blood stage essential genes by genomic deletion. We attempted to disrupt AP2-EXP protein abundance with the conditional knockdown approaches Knock Sideways [49], TetR:DOZI mRNA repression [61], and *glms* ribozyme mediated cleavage [62]. Each of these genetic systems failed to mediate protein knockdown or mislocalization (**S9A, S9B, S22 and S23A–S23C Figs and S1 Table**).

To account for this limitation, we tested the ApiAP2 competitor Compounds B and C against rodent infectious *P. berghei* parasites due to the well-characterized genetic phenotype of the AP2-EXP orthologue PbAP2-Sp [30,55] as a master regulator of sporogenesis. PbAP2-Sp and AP2-EXP are highly similar at amino acid level, and the DNA contacting amino acids identified *via* crystallographic characterization of AP2-EXP [40] are conserved between

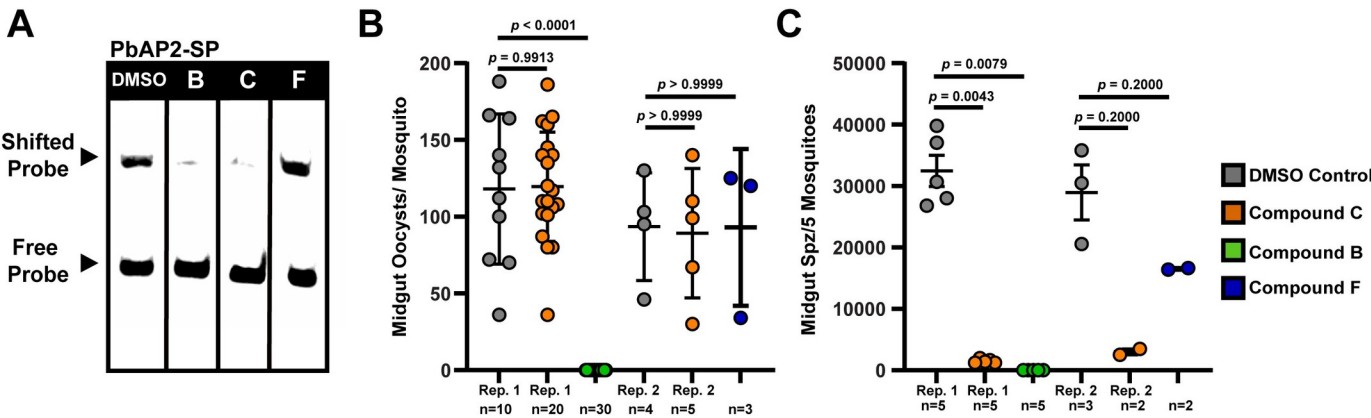

**Fig 5. Compounds B and C are active against mosquito stage *P. berghei* parasites.** A) Compounds B, C and F were added to an EMSA with PbAP2-Sp. Activity against PbAP2-Sp will result in the loss of the shifted probe. 150 fmoles of DNA probe, 125ng of PbAP2-Sp, and 300μM of each compound were used for each lane. DMSO vehicle was used as a control for normal DNA binding by PbAP2-Sp. B) Midgut oocyst counts per mosquito for *P. berghei* infected mosquitoes on day 14 post infection following compound injection on day 10 post infection. Compound identity is indicated in the legend. Rep. 1 and Rep. 2 correspond to biological replicates one and two, respectively. Significance of difference between DMSO vehicle injected samples and compound injected samples was assessed using the Mann Whitney U-test. Bars indicate which compound injected dataset has been compared to the DMSO control dataset. Error bars represent the standard deviation of the mean. C) Midgut sporozoite counts per five infected mosquitoes on day 14 post infection following compound injection on day 10 post infection. Compound identity is indicated in the legend. Rep. 1 and Rep. 2 correspond to biological replicates one and two, respectively. Significance of difference between DMSO vehicle injected samples and compound injected samples was assessed using the Mann Whitney U-test. Bars indicate which compound injected dataset has been compared to the DMSO control dataset. Error bars represent the standard deviation of the mean.

AP2-EXP and PbAP2-Sp (**S24 Fig**). However, to ensure that PbAP2-Sp DNA binding is similarly inhibited in a competitive EMSA, we first tested the ability of Compounds B, C, and F to prevent PbAP2-Sp from binding DNA. As with AP2-EXP, Compounds B and C block PbAP2-Sp DNA binding *in vitro*, while Compound F had no effect (**Fig 5A**).

Since these compounds compete PbAP2-Sp *in vitro*, we injected Compound B or C into the midgut of *Anopheles* mosquitoes infected with *P. berghei* parasites to determine their effect on *Plasmodium* mosquito stage development (**S25 Fig**). Compound F was used as a non-AP2 domain inhibiting control. Mosquitoes injected with Compounds C or F developed comparable oocyst numbers to the DMSO control (**Fig 5B**) while Compound B prevented the development of midgut oocysts entirely (**Fig 5B**). Compound C treated mosquitoes developed 10-fold fewer midgut sporozoites than the control (**Fig 5C**). Therefore, consistent with their *in vitro* activity against ApiAP2 proteins, Compounds B and C are strong inhibitors of *P. berghei* sporozoite development *in vivo* (parasite counts in **S5 Table**).

## Discussion

ApiAP2 transcription factors are unique to Apicomplexan parasites due to their plant-like domain architecture [13,14] and many are essential to asexual blood stage development, making them valuable as potential drug targets. It is therefore desirable to discover chemical scaffolds that can target essential ApiAP2 proteins. Furthermore, chemical inhibition of ApiAP2 proteins may be used in future studies to uncover details about their biological functions.

Using a combination of *in silico*, biochemical, and genetic approaches, we characterized compounds (Compounds A, B, C, and I) that prevent DNA binding by ApiAP2 proteins *in vitro*. Compounds A, B, and C vary in their measured $IC_{50}$, indicating that ApiAP2 inhibiting compounds may be refined to have increased efficacy against *Plasmodium* parasites. Furthermore, Compounds B and C can inhibit *P. berghei* parasite development without killing mosquitoes, suggesting that ApiAP2 competitor compounds may be tolerated by the host and

could serve as transmission blocking agents [63]. Due to their inclusion in the TCAMS, Compounds B, B-1, B-2, B-3, and B-4 have all been tested for activity against the human HEPG2 cell line [42]. ApiAP2 competitor Compounds B and B-4 inhibit HEPG2 cell growth by just 4 and 8%, respectively [42]. Therefore, either drug may potentially be prioritized for further development based on selectivity for *Plasmodium* parasites. Our present results cannot rule out whether Compounds B and C have additional off-target effects, however, a recent series of studies demonstrated the potential to improve a small molecule targeted against the STAT3 DNA binding domain from a lead candidate to reduce off-target effects [39,64], providing a precedent that the compounds described herein could be refined to have greater specificity for the ApiAP2 DNA binding interface in the future.

Compounds B and C have no activity against the plant encoded AtERF1 or human encoded SOX2 proteins, suggesting that they have selectivity for Apicomplexan AP2 domains relative to the other compounds tested. Although the mode of DNA binding is shared between the AP2 domains of AP2-EXP and AtERF1 [44], the specific amino acids that contact DNA are not strictly conserved [40]. Therefore, the lack of activity against AtERF1 may be the result of filtering molecular docking hits based on proximity to DNA base contacting amino acids. Since it is the only ApiAP2 inhibiting compound identified with highly similar analogues available, we used Compound B to make inferences regarding the determinants of activity against AP2 domain proteins based on substitutions to the benzene ring. In a molecular dynamics simulation, Compound B remains stably associated with AP2-EXP, while the non ApiAP2 competitor Compound B-3 does not. This suggests that a steric clash between AP2-EXP and the bromine atom on Compound B-3 is responsible for its lack of DNA binding competition. Despite the lack of complete agreement between the molecular dynamics simulations and competitive EMSAs, this simulation provides a preliminary insight into the lower efficacy of DNA binding competition observed for Compounds B-1, B-2, and B-3 compared to B and B-4. As the competitive EMSA is able to differentiate between Compound C and antimalarials which do not target ApiAP2 proteins (**S5G Fig**), it is possible that Compounds B, B-1, and B-4 have different targets from Compounds B-2 and B-3 *in vivo* despite their similar structures. In the future, testing putative ApiAP2 inhibitors against the full panel of 27 ApiAP2 proteins encoded by *P. falciparum* may further elucidate whether these compounds have different ApiAP2 targets as well.

Based on the description of its genome-wide DNA binding sites and transcript dysregulation in the presence of Compound C, new inferences can be made about the biological function of AP2-EXP. This demonstrates the utility of combining chemical and traditional genetics, because previous attempts to genetically characterize AP2-EXP have not provided a complete picture of its function in the asexual blood stage. The *ap2-exp* AP2 domain coding region has previously been shown to be essential for the asexual blood stage by both saturating mutagenesis and targeted deletion attempts [15–17,21]. Conversely, the coding region beyond the AP2 domain was truncated in two studies [15,21]. Transcriptomic profiling of a truncated AP2-EXP parasite line revealed that many transcripts encoding exported proteins were dysregulated [21]. Unexpectedly, AP2-EXP target genes identified in this study overlap poorly with the differentially regulated transcripts reported as a result of truncation of AP2-EXP (**S4 Table**) [21]. There is greater overlap between our study and AP2-EXP ChIP-seq recently described by Shang *et al* [17], with 50/101 target genes conserved (**S4 Table**). This may explain the apparent essentiality of the AP2 domain, while the full-length AP2-EXP has roles that impact a different subset of non-essential genes.

Since Compound C prevents DNA binding by AP2-EXP, AP2-I D3 and PfSIP2 D1 (75% of tested ApiAP2 proteins) *in vitro*, it is possible that the phenotypes measured arise from multi-ApiAP2 inhibition. A comparison of differentially abundant transcripts in the presence of

Compound C to the target genes of AP2-I [18] revealed that 54/85 AP2-I target genes are decreased in abundance by $\log_2$ fold change >2 at 24–30 hpi. Interestingly, 22/54 of these are also AP2-EXP targets, implying the potential for co-regulation of certain gene subsets. This overlap is consistent with our EMSA results and corroborates the hypothesis that Compound C inhibits both AP2-EXP and AP2-I. Only 3/22 PfSIP2 target genes are decreased in abundance in the presence of Compound C (**ApiAP2 Target Gene Results Summarized in S3 Table**). This may reflect the cryptic relationship between PfSIP2 DNA binding and transcriptional control, as was noted when PfSIP2 was originally characterized as playing a role in genome integrity and heterochromatin formation [20]. Recent work published by Shang *et al*. has provided ChIP-seq datasets for thirteen ApiAP2 proteins [17]. A comprehensive comparison of transcript dysregulation caused by Compound C to each ApiAP2 protein revealed a range (0–22%) of overlap between differential gene abundance and ApiAP2 target genes (**S3 Table**). Seven ApiAP2 proteins (PF3D7_0613800, PF3D7_0802100, PF3D7_1107800, AP2-G5, AP2-HS, AP2-L, AP2-O5) have a >10% overlap between genes decreased in abundance by $\log_2$ fold change >2 and their target genes (**S3 Table**), providing evidence that Compound C may have multi-ApiAP2 inhibiting activity *in vivo*. AP2-EXP and AP2-I have the greatest proportion of dysregulated target genes amongst the characterized ApiAP2 proteins at 50% and 64%, respectively. Possible pan-ApiAP2 inhibition may prove to be a desired goal for future development.

Several ApiAP2 proteins have been implicated in oocyst and sporozoite development in a *P. berghei* ApiAP2 knockout screen [24]. Since Compounds B and C inhibit mosquito stage development of *P. berghei* with differing phenotypes, it is possible that they target different ApiAP2 proteins *in vivo*. Since PbAP2-Sp is required for sporogony, Compounds B and C should minimally inhibit sporozoite development, which is consistent with our results. Overall, these data support a potential for multi-AP2 domain and multi life stage activity for Compounds B and C.

AP2-EXP remains the only ApiAP2 protein for which a structure has been solved. Surprisingly, over a decade later, the field still lacks a clear understanding of the biological role of AP2-EXP or insights into the druggability of the AP2 domain. Our study has enabled us to make inferences about the role of AP2-EXP and set a proof of principle for targeting the highly unique Apicomplexan AP2 DNA binding proteins as a new antimalarial strategy.

## Materials and methods

### Ethics statement

The human red blood cell samples were sourced ethically and their research use was in accord with the terms of the informed consents under an IRB/EC approved protocol (Protocol title: Prospective Collection of Biological Specimens from Subjects Presenting at Specimen Donation Centers for Research, IRB Tracking Number: 20161665, Sponsor: SERATRIALS, LLC). Written consent was obtained from the participants.

All *in vivo* studies using laboratory mice were conducted in accordance with the GSK Policy on the Care, Welfare and Treatment of Laboratory Animals and were reviewed by the ethical review process at the institution where the work was performed. Mice were anesthetized using intraperitoneal injection of 150µl of the following mixture: {2 ml of 100 mg/ml Ketaject (Phoenix Pharmaceutical) + 1 ml of 10 mg/ml ACE Promazine (Henry Schein) + 7 ml saline}. Toe pinching was used to check that the mice were appropriately anesthetized. Protocols were approved by the Johns Hopkins University Animal Care and Use Committee (ACUC, Protocol Number: M018H18).

## Parasite lines

Parasites were grown at 37˚C, 5% O2, 7% CO2 using RPMI1640 media supplemented with hypoxanthine and 0.5% Albumax II (Thermo). The parasite lines used in this study were AP2-EXP::GFP, AP2-EXP::HA, AP2-EXP::*glms*, and wild type Pf3D7 (Malaria Research and Reference Reagents Repository). The human biological samples were sourced ethically and their research use was in accord with the terms of the informed consents under an IRB/EC approved protocol (Protocol title: Prospective Collection of Biological Specimens from Subjects Presenting at Specimen Donation Centers for Research, IRB Tracking Number: 20161665, Sponsor: SERATRIALS, LLC). Written consent was obtained from the participants.

To create AP2-EXP::GFP the C-Terminal coding region of AP2-EXP was cloned (**S1 Table**) into the plasmid pSLI::2xFKBP [49] for endogenous tagging by single homologous recombination (**S9A Fig**). Transgenic parasites with the correct C-Terminal tag were further transfected with pLyn-FRB-mCherry [49] for inducible mislocalization using the same method. Parasites were maintained in media with 2.5nM WR99210. All ChIP and western blot experiments using AP2-EXP::GFP except **S23C Fig** were collected using AP2-EXP::GFP without the pLyn mislocalization plasmid.

To create AP2-EXP::HA we used the pSN054 vector system [65]. The right homology region (RHR) was amplified by PCR (**S1 Table**), and the recodonized left homology region (LHR) was synthesized. Synthesized single guide RNA fragments (**S1 Table**) were cloned into the linearized pSN054 donor vector [65] The parental parasite line used for transfection expresses Cas9 and T7 RNA polymerase [66] (**S9B Fig**). Cell cultures were maintained in 500nM anhydrotetracycline (aTc, Sigma-Aldrich 37919) and 2.5 mg/mL of Blasticidin S.

To create AP2-EXP::*glms*::HA, the C-terminal coding region of AP2-EXP (**S1 Table**) was cloned into the plasmid pSLI::3xHA::*glms* [67] (**S22 Fig**). Parasite cultures were maintained in media with 2.5nM WR99210.

All *P. falciparum* transfections were performed as described [49,68]. All parasite strains were cloned by limiting dilution and genotyped using PCR. AP2-EXP::GFP and AP2-EXP::HA were also genotyped using whole genome sequencing (**NCBI SRA: PRJNA818769: www.ncbi.nlm.nih.gov/bioproject/?term=PRJNA818769**)

## Genomic DNA isolation

Parasite cultures were lysed with 0.1% Saponin in 1xPBS and collected by centrifugation at 1500 RPM, then resuspended in 1xPBS. Genomic DNA was then isolated using the Qiagen DNeasy nucleic acid isolation kit according to the manufacturer's instructions.

## AP2-EXP knockdown assays

For AP2-EXP::GFP + pLyn, synchronous parasites were split into two populations. 250nM Rapalog was added to one parasite group for 48 hours as described [49] while the second group was used as a control. After 48 hours parasite protein was collected in nuclear and cytosolic fractions and AP2-EXP::GFP, Histone H3, and Pf Aldolase were detected by western blot.

For AP2-EXP::HA, parasites were grown routinely in the presence of 500nM aTc. aTc was washed out of one group, while the second group was maintained with aTc as a control for 120 hours before protein harvest. AP2-EXP::HA and Histone H3 were detected by western blot.

For AP2-EXP::*glms*::HA, synchronous parasites were split into two populations. 5mM glucosamine was added to one parasite group for 72 hours as described [62] while the second group was used as a control. After 72 hours parasite protein was harvested for detection of AP2-EXP::*glms*::HA and Histone H3 by western blot.

## *In silico* docking screen

The Tres Cantos Antimalarial Set of 13533 small molecules was retrieved from the supplementary material of Gamo *et al* [42] and downloaded in Spatial Data File (SDF) format from Batch-Entrez. 4603 small molecule structures were downloaded from DrugBank [43] version 2.5 in SDF format. Using the SDF library, a 3-dimensional MOL2 library was created with the use of BALLOON and the Merck Molecular Force Field. BALLOON was used to calculate up to 20 conformations using a genetic algorithm and to select the conformation with the lowest energy of conformation. The 3-dimensional MOL2 library was then prepared for AutoDock using the python script "ligand_prepare4.py" from AutoDockTools [69]. The prepared ligands were then saved in PDBQT file format. The python script "ligand_prepare4.py" automatically defines the torsions of the molecule.

The homodimerized AP2-EXP AP2 domain was downloaded from the solved crystal structure (PDB ID 3IGM). AutoDockTools was used to remove water molecules, add hydrogens, and calculate charges. Residues Arg-88 and Asn-118 were found to be flexible using FlexPred and set as flexible residues for docking. The flexible residues were saved as flexible residues in a flexible residue file and the remaining molecule was saved as rigid residues in a rigid residue file as specified in the AutoDock manual [69]. The rigid residues file was used to pre-calculate an energy grid of the macromolecule using autogrid4 [69]. A set of 14 grid map types S, Cl, F, A, Br, N, P, OA, SA, C, I, HD, Ca, NA were calculated for the macromolecule using AutoGrid [69].

From the prepared ligands a docking set was prepared for AutoDock with the macromolecule and pre-calculated energy grids. For each docking set a docking parameter file was prepared, with the Lamarckian Genetic Algorithm (LGA) specified as the search algorithm for AutoDock. To allow for high throughput parallel processing PERL was used on top of Auto-Dock to manage the dockings. The algorithm was initially set to run with a maximum of 25000 energy evaluations and 20 repeats. Since the calculations are computationally intensive, this setting was used as initial screening of the ligands, to identify a subset of ligands for a more thorough evaluation [69]. The 1000 best hits from the GSK compound evaluation and the DrugBank compound evaluation were selected for a more thorough evaluation. The docking parameters were reconfigured to 250000 energy evaluations and 100 repeats and docking repeated. The docking results were examined with PERL, to provide an automated approach to the interpretation of the docking results. The docking results were evaluated using two different approaches. The top candidates for competition of DNA were selected using a PERL script to filter based on having a geometric center within 10 Å of the location of sense-strand DNA binding. Hits were then filtered based on having a predicted free energy of interaction < -5 kJ/Mol.

## IC$_{50}$ determination

The IC$_{50}$ of each putative ApiAP2 competing compound was determined using a 48-hour growth inhibition assay as described with minor modifications [70]. Briefly, parasites were seeded at 0.5% parasitemia and 4% hematocrit on a 96 well plate. Serial dilutions of drug or DMSO vehicle control were added in to each well and parasites were incubated in standard culture conditions for 48 hours. The assay was run in biological triplicate. Parasite cultures were frozen at -80°C overnight, then 100μL of culture was added to 100μL 2x Lysis Buffer containing 0.2μL/1mL Sybr Green I (20 mM Tris pH 7.5, 5 mM EDTA, 0.008% saponin, 0.08% Triton X-100). The mixture was incubated at room temperature for three hours, then fluorescence was measured at excitation/emission 585nm/535nm for each well. Growth values were normalized to the vehicle control. IC$_{50}$ values were calculated and growth inhibition curves

were plotted using GraphPad Prism with the following parameters: nonlinear regression, Dose-Response—Special, X is log(concentration).

## Recombinant protein expression

Recombinant AP2 domains AP2-EXP, AP2-I D3, AP2-HS D1, and PfSIP2 D1 were overexpressed and purified from BL21 PlysS *E. coli* as described previously [12]. Recombinant AtERF1 was purified using the same method after cloning the AtERF1 AP2 domain [44] into the pGex4t-1 overexpression vector (**S1 Table**). The PbAP2-Sp coding sequence was cloned into the pGex4t-1 overexpression vector using the domain boundaries defined previously (**S1 Table**) [29]. Recombinant protein was quantified using Braford reagent (Pierce). Input, flowthrough, and eluate fractions were analyzed by SDS-PAGE to ensure recovery of the full-length recombinant protein. Recombinant full-length SOX2 protein was purchased from Abcam (ab169843).

## Electrophoretic mobility shift assays

EMSAs were run in DNA binding buffer (10mM Tris pH 7.5, 50mM KCl, 1mM DTT, 6mM MgCl2, 60ng/μL Poly DiDC, 65ng BSA). Recombinant proteins were titrated to empirically determine the minimum mass required for DNA binding and this mass (**S4 Fig**) was used for each EMSA unless otherwise specified. PAGE purified DNA probes with a 5' biotin ligated on the forward DNA strand, along with an unlabeled complementary stand, were purchased from IDT (**S1 Table**). DNA probes were double stranded by heating to 95˚C, followed by stepwise cooling in Annealing Buffer (10mMTtris-Cl pH7.5, 1mM EDTA, 10mM NaCl). Each recombinant protein was incubated in DNA binding mixture plus competitor compound for 15 minutes prior to addition of the cognate double stranded DNA oligonucleotide. Protein, competitor, and DNA oligonucleotide were incubated together for an additional 5 minutes. The mixtures were separated on a .5x TBE polyacrylamide gel, transferred to a nylon membrane (Amersham) at 50 Volts for 30 minutes, and probed using the Light Shift nucleic acid detection module (Thermo) according to the manufacturer's protocol. All gels were imaged using a Bio Rad chemiluminescence imager.

## Ethidium bromide exclusion assay

10μM of double stranded DNA (sequence TGCATGCA, purchased from IDT) in 0.01mM EDTA, 9.4mM NaCl, 2mM HEPES buffer pH 7.9 was incubated for 10 minutes in the presence of 100nM ethidium bromide. Fluorescence was measured at excitation/emission 546nm/595nm for each well. Following the baseline reading, each putative ApiAP2 competing compound was added. DRAQ5 nucleic acid dye was used as a positive control for knockdown of ethidium bromide fluorescence by DNA intercalation. Ethidium bromide exclusion assays were performed in technical triplicate using a 96 well plate.

## Molecular dynamics simulations

Five compounds were prepared for molecular dynamics (MD) simulation: B, B-1, B-2, B-3, and B-4. In preparation for MD, parameters for the compounds were obtained using Antechamber [71–73] with the Generalized Amber Force Field. Starting conformations for each compound bound to AP2-EXP were based on predictions from docking. All complexes were prepared using the tleap module of AmberTools [74] with the protein.ff14SB forcefield [75]. Each complex was solvated in an octahedral box of TIP3P water with a 10 -Å buffer around the protein complex. Na$^+$ and Cl$^-$ ions were added to neutralize the protein and achieve

physiological conditions. All MD minimizations and simulations were performed using Amber with GPU acceleration [76,77]. First, complexes minimized with 5000 steps each of steepest decent and conjugate gradient minimization with 500 kcal/mol·$\text{Å}^2$ restraints on all complex atoms. Restraints were reduced to 100 kcal/mol·$\text{Å}^2$ and the minimization protocol was repeated. Restraints were then retained only on the compound for a final minimization step. Following minimization, all complexes were heated from 0 to 300 K using a 100-ps run with constant volume periodic boundaries and 10 kcal/mol·$\text{Å}^2$ restraints on all protein and compound atoms. To equilibrate complexes, 10 ns of MD was performed first with 10 kcal/mol·$\text{Å}^2$, then with 1 kcal/mol·$\text{Å}^2$ restraints on protein and compound atoms using the NPT ensemble. With kcal/mol·$\text{Å}^2$ restraints retained on complexes, 500 ns production simulations were performed. A 2-fs timestep was used and all bonds between heavy atoms and hydrogens were fixed with the SHAKE algorithm [78]. A cut-off distance of 10 Å was used to evaluate long-range electrostatics with Particle Mesh Ewald (PME) and for van der Waals forces. The 'strip' and 'trajout' commands of the CPPTRAJ module [79] were used to remove solvent atoms and extract 50,000 evenly spaced frames from each simulation for analysis.

## AP2 domain competitor phenotyping assays

For Compound B, C, and H phenotyping time courses (**Fig 3A-3E**), highly synchronous Pf3D7 wild type parasites at 10% starting parasitemia were spiked with 40μM of each drug at 6 hpi. An equivalent volume of DMSO was used as a control. Parasites were morphologically assessed by Giemsa staining and parasitemia was counted every 6 hours until 48 hpi.

For the Compound C exposure interval assay (**Figs 3F and S8A**) 40μM of Compound C was spiked into highly synchronous Pf3D7 wild type parasites at 1% starting parasitemia for individual 8-hour intervals starting at 6 hpi. An equivalent volume of DMSO vehicle was used as a control for normal growth. At 54 hpi, cultures were counted using Giemsa-stained slides (Raw data in **S8B Fig**). In **Fig 3F**, Compound C induced growth inhibition is reported at the ratio of nascent rings to stalled trophozoites in culture at 54 hpi. The Mann Whitney U-test was used to check for significance of difference between each dosage interval and the DMSO vehicle control.

To determine the phenotype caused by continuous exposure of 0.66x$\text{IC}_{50}$ Compound C to wild type Pf3D7 parasites in **S13A, and S13B Fig,** highly synchronous Pf3D7 parasites at 5% starting parasitemia were spiked with 12μM Compound C, 40μM Compound C, or DMSO vehicle control at 10 hpi. The percentages of mature trophozoites and reinvaded rings were recorded at 24 hpi and 48 hpi, respectively.

For all phenotyping assays, 40μM of compound was used because it was the maximum concentration achievable in culture after resuspending each compound in 100% DMSO. Higher concentrations of compound caused parasite mortality due to the high volume of DMSO required (>1% volume/volume). All assays were performed in biological and technical triplicate. Error bars represent the standard deviation of the mean.

## Preparation of RNA for DNA microarray

DNA microarrays were prepared using the protocol described in [50]. After collecting a control timepoint at 6 hpi, parasites were cultured with either 12μM Compound C (0.66x$\text{IC}_{50}$) or DMSO vehicle control. Parasite RNA was then harvested at 12, 18, 24, 30, 42, and 48 hpi. cDNA was synthesized using SuperScript II, hybridized onto Agilent DNA microarrays [50], and scanned using an Axon 4200A Scanner. Agilent Feature Extraction Software version 11.0.1.1 with the protocol GE2-v5_95_Feb07_nospikein was used to extract signal intensities. Raw signal intensities for all microarrays are available in NCBI GEO **GSE202876**: https://www.ncbi.nlm.nih.gov/geo/query/acc.cgi?acc=GSE202876

## Analysis of DNA microarrays

LIMMA [52] was used to normalize and extract signal intensities. Arrays were normalized using robust splines normalization and within arrays parameters selected. Average expression values were calculated per gene by averaging the $\log_2$ Cy5 (cDNA)/Cy3 (Reference Pool) signal intensity across all probes. Correlation plots for the Compound C vs. DMSO control total transcriptome and control genes were made using the $\log_2$ signal intensities for all detected genes using the R package CorrPlot downloaded from: https://github.com/taiyun/corrplot. The significance of correlation was tested using the cor.mtest function of CorrPlot with confidence level = 0.95. The LIMMA eBayes function was used on average gene abundance data to determine changes in transcript abundance between 24 and 30hpi that occur differentially in DMSO control vs. Compound C dosed parasites. Volcano plots were made with LIMMA eBayes data using the Enhanced Volcano package in R downloaded from: https://github.com/kevinblighe/EnhancedVolcano. The Rnits [54] R package was used to model differences in gene expression across the entire time course between the DMSO control and Compound C spike in parasites. Rnits was run using the parameters: center genes, normalize by intensity, and background filter probes. The Rnits model for differential expression was fit at the gene level. All data from LIMMA, eBayes, and Rnits is provided in **S3 Table**. Heat maps for control genes and AP2-EXP target genes were made using Java Treeview [80] after means centering of transcript abundance values with Cluster3.0 [81].

## ChIP sample preparation

ChIP-seq was performed as described in [19]. Highly synchronous parasites were grown up to 5–10% parasitemia and AP2-EXP ChIP was performed at 30 hpi for all replicates. Immunoprecipitation was performed overnight with .5mg/mL 3F10 anti-HA (Sigma) or .1mg/mL Ab290 (Abcam) anti-GFP.

## DNA library preparation

ChIP-seq DNA libraries were prepared as described in [19] using the NEBNext II DNA library kit (New England Biolabs) according to the manufacturer's instructions. Quality was assessed using an Agilent 2100 Bioanalyzer or TapeStation. Libraries were sequenced using a HiSeq 2500 (Replicate GFP1) or NextSeq 550 (Replicates GFP2 and HA3) Illumina sequencer. AMPure XP beads (Beckmann Coulter) were used to size select and purify DNA between NEBNext II library preparation steps. Whole genome sequencing DNA libraries were prepared using the Illumina Tru-Seq PCR free DNA library kit and sequenced using a HiSeq 2500.

## qPCR

ChIP-qPCR samples were collected from 30–35 hpi trophozoites following two hours of Compound C (40μM) or DMSO vehicle control spike in. Primer pairs to be used for ChIP-qPCR were first evaluated to check for 80–110% efficiency using sonicated genomic DNA. RT-qPCR was carried out using Sybr Green Polymerase master mix (Thermo) with the specified primer concentration (**S1 Table**). The Ct was calculated using SDSv1.4 (Applied Biosystems) software, averaged over technical triplicate. The percent of input per immunoprecipitated DNA fraction was calculated using the delta Ct method. Each assay was performed in biological triplicate, with the exception of the *ama1* primer pair, where n = 2. Error bars represent standard deviation of the mean. Data was obtained using an Applied Biosystems 7300 Real-Time PCR Machine. Significance was assessed using an unpaired t-test.

## ChIP-seq data analysis

First, reads in each DNA library were trimmed to remove low quality base calls and Illumina adaptor sequences using Trimmomatic [82] with a quality score cutoff of 20. FastQC [83] was used to assess the quality of each DNA library following this step. Reads were then mapped to the *Plasmodium falciparum* genome version 36 downloaded from PlasmoDB using BWA-Mem [84]. Multiply mapped reads were filtered out using Samtools [85]. Filtered and mapped Input and immunoprecipitate bam files were used for peak calling by MACS2 [86] with the following parameters: effective genome size 20000000, No Model, q 0.01. Peaks called in each ChIP-seq replicate were overlapped using Bedtools [87] Intersect, and peaks that occur in a minimum of 2/3 replicates were used for downstream analysis. For data visualization using IGV, .bam files were converted to $\log_2$ Immunoprecipitate fraction/Input fraction bigwigs using DeepTools [88] BamCompare. Bedtools [87] ClosestBed was used to correlate regions called as peaks of occupancy by MACS2 to the closest gene. Results were filtered based on 1.5kb proximity to the MACS2 peak and strandedness of the putative target gene. For comparison of ChIP-seq library coverages to Transcription Start Sites and Start Codons, DeepTools PlotHeatmap was used. Matrix files underlying PlotHeatmap were created using DeepTools ComputeMatrix. A four-color plot of conserved AP2-EXP DNA binding sites was made using the program cegr-tools four color plot downloaded from https://github.com/seqcode/cegr-tools/tree/master/src/org/seqcode/cegrtools. DNA motifs enriched in AP2-EXP ChIP-seq peaks were determined using the DREME Suite [89]. DNA motifs were sorted from highest to lowest level of conservation to the aggregate AP2-EXP DNA motif using FIMO [90].

For comparison of AP2-EXP peaks with existing post-translational histone marks, chromatin post translational modification datasets [57,91] were downloaded from https://github.com/Daread/plasmodiumExprPrediction [92]. Chromatin reader and nucleosome occupancy datasets [58,93,94] were downloaded from NCBI. ChIP-seq data for BDP1 and HP1 was processed as described above. Nucleosome occupancy data was downloaded from NCBI and processed as described above, with the exception that bam files were turned into bigwig using BamtoBigWig with the—mnase parameter selected. DeepTools ComputeMatrix was used to format the data. DeepTools PlotHeatmap was used to plot coverage of each dataset within 10kb of AP2-EXP peaks. For negative control coverage plots, Bedtools [87] ShuffleBed was used to create random genomic intervals on the same chromosome and the same length as AP2-EXP peaks of occupancy.

For comparison of AP2-EXP peaks identified by Shang *et al* [17] to this study, AP2-EXP peaks of occupancy in the trophozoite stage were downloaded from https://www.ncbi.nlm.nih.gov/geo/query/acc.cgi?acc=GSE184658. Peak summits from each replicate were expanded to 200 nucleotide windows using Bedtools SlopBed [87]. Conserved peaks in 2/2 replicates were identified using Bedtools intersect [87], then AP2-EXP peaks from this study and Shang *et al* were overlapped using Bedtools Intersect [87].

## Western blot

Full parasite protein western blot samples were collected by lysing RBCs with 0.1% saponin and boiling protein in Loading Buffer (50mM Tris-Cl pH 8.0, 20% SDS, 1% Bromophenol Blue). Fractionated parasite protein was prepared as described in [95]. Blots were performed as described [19]. Primary antibodies used were: 1/1000 rat ant-HA (Roche 3F10), 1/1000 mouse anti-GFP (Roche), 1/3000 rabbit anti-aldolase conjugated to HRP (Abcam ab38905), or 1/3000 mouse anti-H3 (Abcam ab10799). Secondary antibody concentrations used were 1/3000 goat anti-rat HRP conjugate (Millipore), 1/3000 goat anti-mouse HRP conjugate, or 1/10,000 (Pierce) goat anti-rabbit HRP conjugate (Millipore). ECL reagent (Pierce) was used to

detect HRP signal. Blots were exposed to autoradiography film (VWR) and visualized using an autoradiography developer.

## Protein pulldown

Parasite nuclear protein was isolated as described [95]. Following the final centrifugation step, the supernatant was collected and diluted by 1:3 in Dilution Buffer (30% glycerol, 20mM HEPES pH 7.8). GFP tagged AP2-EXP was pulled down using Chromotek anti-GFP or mock immunoprecipitated using Chromotek negative control magnetic beads. Beads were washed twice in Wash Buffer prior to use (20mM HEPES pH 7.4, 250mM NaCl, 1mM EDTA, 1mM TCEP, 0.05% NP-40). Protein and beads were incubated together for 1 hour at 4˚C. The beads were then washed twice with Wash Buffer and bound protein was collected in Loading Buffer by boiling at 95˚C for 10 minutes.

## Fluorescent microscopy

Samples were prepared by incubating packed infected red blood cells with DRAQ5 dye (Thermo) for 15 minutes. Parasites were then washed in 1xPBS to remove excess dye and immediately placed on a glass slide for imaging. Fluorescent microscopy images were acquired using an Olympus Bx61 fluorescent microscope. All images were processed using SlideBook 5.0.

## Mosquito stage *P. berghei* AP2 competitor growth assay

A female Swiss Webster mouse was inoculated with Plasmodium berghei ANKA 2.34 from frozen stock. Once the parasitemia reached 15%, the blood was harvested by heart puncture, washed twice with 1xPBS and resuspended to 10 mL in 1xPBS. 5 female Swiss Webster mice were infected with 500µL of the resuspended blood. 3–4 days after blood passage, exflagellation of the *P. berghei* gametocytes was assayed. Briefly, a drop of tail vein blood was incubated in RPMI 1640 (Invitrogen) containing 1µM xanthurenic acid (Sigma) for ~12–15 minutes. The mixture was added to a slide and observed under a light microscope at 40x. Ten or more fields were observed. Mice with 0.3–0.7 exflagellations per field were anesthetized and fed on 3-day post emergence *Anopheles stephensi* mosquitoes. Mosquitoes were maintained at 19˚C and fed 10% sucrose. Bloodfed mosquitoes were separated 30 hours post-bloodfeeding. 10 days post-bloodfeeding, 5–10 mosquitoes were dissected and oocysts were counted by mercurochrome staining and light microscopy to ensure *P. berghei* oocyst development. The remaining mosquitoes were injected by standard mouth pipette technique with small molecule inhibitors (18µM Compound C, 40µM Compound B, 20µM Compound F) or a mixture of PBS-DMSO for control. The surviving injected mosquitoes were dissected 14 days post-bloodfeeding. Oocyst numbers were counted as before. Pooled groups of midguts were ground using a pestle, centrifuged at 7,000 rpm for 5 minutes and resuspended in 20µL 1xPBS to release developing mid-gut sporozoites. Sporozoites were counted on a haemocytometer. Significance of difference between DMSO control and compound injected samples was assessed using the Mann Whitney U-test.

## Supporting information

**S1 Data. Excel spreadsheet containing the underlying numerical data for Figs 3A–3D, 3F, S3A–S3I, S7B, S8B, S13A, and S17.**
(XLSX)

**S1 Table. List of oligonucleotides used in this study.**
(XLSX)

**S2 Table. Summary of the Competitive EMSA results for each DNA binding domain tested.**
(XLSX)

**S3 Table. All DNA Microarray expression values.**
(XLSX)

**S4 Table. All AP2-EXP ChIP-seq peaks and gene targets.**
(XLSX)

**S5 Table. *P. berghei* mosquito stage inhibition assay parasite counts.**
(XLSX)

**S1 Movie. Molecular dynamics simulation for Compound B interaction with AP2-EXP.**
(WMV)

**S2 Movie. Molecular dynamics simulation for Compound B-1 interaction with AP2-EXP.**
(WMV)

**S3 Movie. Molecular dynamics simulation for Compound B-2 interaction with AP2-EXP.**
(WMV)

**S4 Movie. Molecular dynamics simulation for Compound B-3 interaction with AP2-EXP.**
(WMV)

**S5 Movie. Molecular dynamics simulation for Compound B-4 interaction with AP2-EXP.**
(WMV)

**S1 Fig. Putative AP2-EXP competitors were identified using computational docking.** A) The crystal structure of AP2-EXP (PDB:3IGM) [40] was used as a template to computationally dock thousands of small molecules *in silico* using AutoDock. Results were filtered for compounds that dock within 10 Angstroms of DNA binding residues with a free energy less than -5kJ/mol. Compounds matching these criteria were sourced and used for further testing. B) Seven compounds were identified as putative ApiAP2 competitors in an *in-silico* screen (Column 2). Six of these were available for direct purchase (Column 3). For the remaining compound (TCMDC-124220), three alternate choices with a Tanimoto similarity score of .9 or greater were purchased (denoted by an asterisk in Column 3). The PubChem ID used to purchase each compound in this study is listed in Column 4.
(TIF)

**S2 Fig. Docking conformations for Compounds A-I.** A) The spatial conformation of each of the nine compounds (A-I) that dock within 10 Angstroms of the DNA binding pocket of AP2-EXP with a free energy less than -5kJ/mol is depicted above. B) Chemical structures of each compound (A-I) corresponding to the molecular docking results in panel A.
(TIF)

**S3 Fig. IC50 assays for Compounds A-I.** A-I) 48-hour Sybr Green growth inhibition assays were conducted for each of the nine putative ApiAP2 competitor compounds in order to determine IC50 values against asexual *P. falciparum*. All growth assays were performed in triplicate. All compounds kill asexual stage *P. falciparum* parasites in the micromolar (11.33–198.90μM) concentration range. Error bars represent standard deviation of the mean.
(TIF)

**S4 Fig. Titration of recombinant DNA binding domains to optimize competition electrophoretic mobility shift (EMSA) assays.** A-F) DNA binding domains AP2-EXP, AP2-I D3, AP2-HS D1, PfSIP2 D1, AtERF1, and full length SOX2 were titrated against DNA oligos containing their respective binding motifs (highlighted in red) in an EMSA. Unless otherwise specified, the minimum mass of each recombinant DNA binding domain required to visualize DNA binding (denoted by an arrow) was used in competitive EMSAs with putative ApiAP2 competitor compounds.
(TIF)

**S5 Fig. Putative ApiAP2 competitor compounds were tested against additional proteins in a competitive EMSA.** A-C) AP2-I D3 (A) DNA binding activity is competed by Compounds A, B, C and I. AP2-HS D1 (B) is competed by Compounds B and I, and PfSIP2 D1 (C) is competed by Compounds B, C, and I. Compounds A, B, C and I all compete at least one AP2 domain in addition to AP2-EXP. Cognate DNA motifs for each protein are highlighted in red. 300μM of each compound was used per lane. D) The plant encoded *Arabidopsis thaliana* AP2 domain from Ethylene Response Factor 1 (AtERF1) is competed by Compound I. The *Plasmodium* AP2 domain competitors Compound A, B, and C do not compete AtERF1. The cognate AtERF1 DNA motif is highlighted in red. 300μM of each compound was used per lane. E) The human encoded High Mobility Group Box Domain transcription factor SOX2 is competed by Compound A. Due to the lack of homology between SOX2 and AP2 domain proteins, this result indicates that Compound A's DNA binding competition activity is not unique to the AP2 domain. The cognate SOX2 DNA motif is highlighted in red. 300μM of each compound was used per lane. F) The three-dimensional structures of AP2-EXP [40] (PDB:3IGM), AtERF1 [44] (PDB: 2GCC), and SOX2 [46] (PDB: 2LE4). AP2-EXP and AtERF1 bind DNA *via* contacts with the beta strands [40], while SOX2 binds DNA *via* contacts with its alpha helices [96]. G) 300μM of the antimalarial compounds pyrimethamine, chloroquine, or artemisinin, were added to a competitive EMSA with AP2-EXP. Compound C was used as a control for activity against AP2-EXP. DMSO was used as a vehicle control. The AP2-EXP cognate DNA motif is highlighted in red. H) The chemical structures of Compound C, pyrimethamine, chloroquine, and artemisinin.
(TIF)

**S6 Fig. Compounds A-I were tested for DNA intercalation in an ethidium bromide exclusion assay.** Each putative ApiAP2 competitor compound was added into a mixture containing double stranded DNA and ethidium bromide. The positive control DNA major groove intercalator DRAQ5 knocks down ethidium bromide fluorescence nearly completely relative to the DNA and ethidium bromide control. The legend indicates the cumulative result for each compound in competitive EMSAs. Each assay was performed in triplicate. Error bars represent standard deviation of the mean.
(TIF)

**S7 Fig. Compound B analogues were tested against AP2-I D3 in a competitive EMSA and checked for DNA intercalation ability.** A) Compounds B, B-1, B-2, B-3, and B-4 were added to an EMSA with AP2-I D3 to check whether their DNA binding competition is consistent with AP2-EXP. The cognate AP2-I D3 DNA motif is highlighted in red. 300μM of each compound was used per lane. B) Compounds B, B-1, B-2, B-3, and B-4 were tested for DNA major groove intercalation in an ethidium bromide exclusion assay. DRAQ5 was used as a positive control for intercalation. Each assay was performed in triplicate. Error bars represent standard deviation of the mean.
(TIF)

**S8 Fig. A fixed interval Compound C dosage assay to determine the timing of antimalarial action.** A) Schematic depicting the phenotyping time course. 40μM Compound C was added to wild type Pf3D7 parasite cultures at 1% starting parasitemia for fixed 8-hour intervals starting at 6 hpi throughout the IDC. The interval of Compound C dosage is indicated by each red bar. B) At 54 hpi following Compound C dose intervals, the parasitemia and morphology of each culture was counted. The total percentage of reinvaded rings and stalled trophozoites in each culture was recorded. The ratio of rings to trophozoites is reported in **Fig 3F**. Error bars represent standard error of the mean. The assay was performed in biological and technical triplicate.
(TIF)

**S9 Fig. Creation of endogenously tagged parasite lines AP2-EXP::GFP and AP2-EXP::HA.** A) The Selection Linked Integration system was used to add 2xFKBP inducible mislocalization protein and GFP to AP2-EXP by single homologous recombination. Successful integration was confirmed by genotyping PCR. Genomic DNA from the wild type Pf3D7 parental line was used as a control. In order to test the efficacy of the knock sideways system, the pLyn mislocalizer plasmid was added to AP2-EXP::GFP and confirmed by PCR. DNA kb are indicated by the marks to the left of the gel. B) The PSN054 TetR:DOZI plasmid was used to add the TetR: DOZI mRNA repression module and endogenous 2xHA tag to AP2-EXP by double homologous recombination. Correct integration was confirmed by genotyping PCR. The parental NF54 parasite line was used as the unedited control. Clonal populations one and two are indicated as C1 and C2, respectively. Clone one had correct integration and complete absence of the wild type *ap2-exp* DNA locus and was used for further experiments. DNA kb are indicated by the marks to the right of the gel.
(TIF)

**S10 Fig. AP2-EXP protein expression in the AP2-EXP::GFP endogenously tagged parasite line (related to Fig 3A).** A) AP2-EXP expression was tracked throughout the IDC by harvesting protein from highly synchronous asexual blood stage parasites followed by a western blot against the GFP tag. Histone H3 was used as a loading control. B) AP2-EXP::GFP expression was monitored in a highly synchronous parasite population by fluorescent microscopy across the IDC. DRAQ5 was used as a nuclear stain for parasites.
(TIF)

**S11 Fig. AP2-EXP protein expression in the AP2-EXP::HA endogenously tagged parasite line.** AP2-EXP expression was tracked throughout the IDC by harvesting protein from highly synchronous asexual blood stage parasites followed by a western blot against the 2xHA tag. Histone H3 was used as a loading control.
(TIF)

**S12 Fig. Quality control of DNA microarray data for DMSO vehicle control and Compound C parasites, related to Fig 4.** A) DNA microarray data from [51] for a set of highly periodic control genes expressed in the IDC. B-C) The same set of highly periodic control genes as in panel A were plotted for the DMSO control and 12μM Compound C spiked parasites in order to compare parasite staging between the two experiments. D) Correlogram depicting the Spearman Correlation value between control gene expression for DMSO (Panel B) and Compound C (Panel C) samples. A $^*$ indicates p value < 0.05. E-F) Correlogram depicting the Spearman Correlation value between the total transcriptome of DMSO control and Compound C dosed parasites. A $^*$ indicates p value < 0.05. In contrast to the minimal perturbation observed in either dataset relative to the control gene set, the total transcriptome is

significantly altered in the Compound C treated parasites compared to the DMSO control.
(TIF)

**S13 Fig. A 48-hour time course to determine the phenotype for 12μM Compound C, related to Fig 4A.** A) Highly synchronous asexual blood stage Pf3D7 parasites were spiked with DMSO vehicle control, 12μM Compound C, or 40μM Compound C. Each growth assay was performed in biological triplicate. Error bars represent standard deviation of the mean. B) Representative images of each parasite population (DMSO, 12μM Compound C, 40μM Compound C) at 24 and 48 hpi.
(TIF)

**S14 Fig. ChIP-seq protein quality control, related to Fig 4B.** A) Crosslinked nuclear material was blotted after sonication to ensure recovery of the full-length AP2-EXP protein during chromatin immunoprecipitation. Full length AP2-EXP is recovered, indicating that the protocol is suitable to analyze AP2-EXP DNA binding *in vivo*. Crosslinked nuclear material from the wildtype Pf3D7 parental parasite line was used as a negative control. B) Anti-GFP beads were used to pull down GFP tagged AP2-EXP from AP2-EXP::GFP. Flowthrough (FT), Wash (W1 and W2) and Eluate fractions were analyzed by western blot. The presence of AP2-EXP and Histone H3 in the Eluate lane indicates that AP2-EXP interacts with chromatin in the nucleus. The non-immune negative control beads do not enrich AP2-EXP or Histone H3 in the eluate.
(TIF)

**S15 Fig. ChIP-seq extended data.** A) Log2 immunoprecipitate/Input ChIP-seq data from each replicate (2x AP2-EXP::GFP and 1xAP2-EXP::HA) of AP2-EXP ChIP-seq visualized by IGV at a representative DNA locus. The location of a conserved MACS2 called peak of occupancy and the TGCATGCA DNA motif is indicated by the bottom tracks. The no epitope control lane is the coverage resulting from a no-epitope control ChIP-seq done using the anti-GFP antibody. B) The top three ranked DNA motifs present within peaks of occupancy for each ChIP-seq replicate as determined by DREME [89]. The core DNA motif CATGCA is overrepresented within each individual replicate. C) The top overrepresented DNA motif within AP2-EXP peaks of occupancy conserved in 2/3 replicates of ChIP-seq as determined by DREME [89] plotted at the primary DNA sequence level. DNA sequences were sorted from highest to lowest degree of motif conservation using FIMO [90].
(TIF)

**S16 Fig. Comparison of AP2-EXP target genes with Compound C induced changes in transcript abundance.** The total overlap between AP2-EXP gene targets detected in the Compound C RNA time course and global decrease in transcript abundance at 24–30 hpi by $\log_2$ fold change >2.
(TIF)

**S17 Fig. ChIP-Quantitative PCR to assess Compound C impact on AP2-EXP genomic occupancy.** AP2-EXP::GFP parasites were spiked with 40μM Compound C or DMSO vehicle control at 30 hpi for two hours. ChIP samples were collected for each population using either anti-GFP or negative control IgG antibodies. The percent of input was determined by RT-qPCR. The presence or absence of an AP2-EXP peak of occupancy at each DNA locus based on ChIP-seq is indicated by a (+) or (-), respectively. Each assay was done in triplicate with the exception of AMAI, where n = 2. Statistical significance was assessed using an unpaired t-test. All results are non-significant (*p*-value >0.05).
(TIF)

**S18 Fig. Nucleosome occupancy is depleted at AP2-EXP DNA binding sites.** Mnase-seq data [93] was plotted against DNA binding sites conserved in 2/3 replicates of AP2-EXP ChIP-seq (A) or random genomic intervals of equal length from the same chromosome as the original peak (B).
(TIF)

**S19 Fig. Histone post translational modifications and chromatin reader occupancy at AP2-EXP peaks.** The occupancy of histone variant H2A.Z, histone modifications H3K9ac, H3K4me3 [57], H3K9me3, H3K36me2/3 [91], and chromatin readers BDP1 [58] and HP1 [94] were plotted against AP2-EXP peaks of occupancy conserved in 2/3 ChIP-seq replicates (A) or random genomic intervals of the same length (B), taken from the same chromosome on which the AP2-EXP peak originally occurred.
(TIF)

**S20 Fig. AP2-EXP DNA occupancy with respect to the Transcription Start Site (TSS) of target genes.** $Log_2$ immunoprecipitate (ChIP)/Input ChIP-seq coverage for each replicate of AP2-EXP ChIP-seq and the no epitope control was plotted against the TSS [59] of each target gene conserved in 2/3 ChIP-seq replicates.
(TIF)

**S21 Fig. Normal Transcript Abundance of AP2-EXP Target Genes.** AP2-EXP target genes were determined by ChIP-seq and their transcript abundance data during the 48-hour IDC was plotted using data from Chappell *et al* [59].
(TIF)

**S22 Fig. Creation of a *glms* ribozyme based knockdown line for AP2-EXP.** Wildtype Pf3D7 *P. falciparum* parasites were transfected to endogenously tag the AP2-EXP DNA locus with an inducible *glms* ribozyme and HA epitope tag. Successful integration to create AP2-EXP::*glms*::HA by single crossover homologous recombination was confirmed by genotyping PCR. C1 and C2 represent clonal populations selected for integration. WR represents a parasite population selected for the plasmid but not for integration. WT represents Pf3D7 wild type control parasite gDNA.
(TIF)

**S23 Fig. Western blot phenotyping of attempts to genetically knockdown AP2-EXP.** A) To assess genetic knockdown of AP2-EXP by *glms* ribozyme tag in the parasite line AP2-EXP::*glms*::HA, highly synchronous parasites were spiked with 5mM glucosamine or vehicle control for 72 hours and AP2-EXP quantity was determined by anti-HA western blot. Histone H3 was used as a loading control. Glucosamine treatment did not impact the amount of AP2-EXP protein present. B) Genetic knockdown of AP2-EXP by the TetR:DOZI mRNA repression module was assessed in the parasite line AP2-EXP::HA by washing anhydrotetracycline (aTc) out of the media for 120 hours. AP2-EXP quantity was determined by anti-HA western blot, and Histone H3 was used as a loading control. Removal of aTc from the media did not impact the amount of AP2-EXP protein present. C) Genetic knockdown of AP2-EXP via protein mislocalization was assessed for the parasite line AP2-EXP::GFP. 250nM rapalog was added to the media for 48 hours and AP2-EXP protein localization was assessed by ant-GFP western blot. Histone H3 and Aldolase were used as nuclear and cytosolic markers, respectively. The addition of rapalog did not cause any detectable mislocalization of AP2-EXP from the nucleus to the cytosol. N indicates the nuclear protein fraction, and C indicates the cytosolic fraction.
(TIF)

**S24 Fig. Sequence alignment between the AP2-EXP and PbAP2-Sp AP2 domains.** A sequence alignment between the AP2 domain of AP2-EXP and PbAP2-Sp. The four amino acids which make base specific contacts with DNA in AP2-EXP are highlighted in red [40]. The domain boundaries of recombinant PbAP2-Sp [30] and AP2-EXP [12] are underlined.
(TIF)

**S25 Fig. Mosquito stage *P. berghei* inhibition assay schematic.** *A. stephensi* mosquitoes were infected with *Plasmodium berghei* parasites. On day 10 post infection, mosquito midguts were injected with Compounds B, C, F, or DMSO vehicle control. On day 14 post infection mosquitoes were dissected to count oocysts and midgut sporozoites. For Compound C and DMSO vehicle control, each experiment was performed in duplicate. Compounds B and F phenotyping were performed as a single experiment.
(TIF)

## Author Contributions

**Conceptualization:** Timothy James Russell, Kathryn Shaw-Saliba, Gianni Panagiotou, Manuel Llinás.

**Data curation:** Timothy James Russell.

**Formal analysis:** Timothy James Russell.

**Funding acquisition:** Manuel Llinás.

**Investigation:** Timothy James Russell, Valerie M. Crowley, Kathryn Shaw-Saliba, Gianni Panagiotou.

**Methodology:** Timothy James Russell, Erandi K. De Silva, Kathryn Shaw-Saliba, Namita Dube, Charisse Flerida A. Pasaje.

**Project administration:** Manuel Llinás.

**Resources:** Erandi K. De Silva, Gabrielle Josling, Charisse Flerida A. Pasaje, Jacquin C. Niles, Marcelo Jacobs-Lorena, Francisco-Javier Gamo.

**Software:** Timothy James Russell, Namita Dube, Irene Kouskoumvekaki, Gianni Panagiotou, C. Denise Okafor.

**Supervision:** Jacquin C. Niles, Marcelo Jacobs-Lorena, C. Denise Okafor, Manuel Llinás.

**Validation:** Timothy James Russell.

**Visualization:** Timothy James Russell, Namita Dube, C. Denise Okafor.

**Writing – original draft:** Timothy James Russell, Erandi K. De Silva, Manuel Llinás.

**Writing – review & editing:** Timothy James Russell, Manuel Llinás.

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
