## [Decision Letter · Decision Letter 0]

30 Jun 2022

Dear Dr. Russell,

Thank you very much for submitting your manuscript "Identification of antimalarial compounds that inhibit Apicomplexan AP2 transcription factor proteins in the human malaria parasite Plasmodium falciparum" for consideration at PLOS Pathogens. As with all papers reviewed by the journal, your manuscript was reviewed by members of the editorial board and by several independent reviewers. In light of the reviews (below this email), we would like to invite the resubmission of a significantly-revised version that takes into account all of the reviewers' comments. Specially all of the reviewers thought that the presented results are unable to convincingly support the assertion in the title that these compounds function by inhibiting DNA-binding in the parasite even at the micromolar concentrations used. They also all thought the presentation of the data was confusing, so you should use their comments to help guide you in a better presentation of the data. 

We cannot make any decision about publication until we have seen the revised manuscript and your response to the reviewers' comments. Your revised manuscript is also likely to be sent to reviewers for further evaluation.

Sincerely,

Laura J. Knoll

Pearls Editor and Guest Associate Editor

PLOS Pathogens

Kami Kim

Section Editor

PLOS Pathogens

Kasturi Haldar

Editor-in-Chief

PLOS Pathogens

orcid.org/0000-0001-5065-158X

Michael Malim

Editor-in-Chief

PLOS Pathogens

orcid.org/0000-0002-7699-2064

Reviewer's Responses to Questions

**Part I - Summary**

Reviewer #1: The manuscript describes in silico identification of a new class of antimalarial compounds targeting the parasite-specific family of transcription factors and their validation both in vivo and in vitro. It identifies some highly promising hits, shows they are binding the proteins of interest and confirms their ability to inhibit parasite growth. As a follow-up the authors use one of these compounds to investigate its putative target -AP2-EXP, and the role it plays in the parasite maturation and gene expression regulation.

The work is very well designed, clearly presented and data-rich. It contains a very thorough validation of the antiparasitic activity of the studied compounds (authors use two parasite species, multiple life stages etc.). There is no doubt it will be of high interest to the parasitology community, especially as the apiAP2 proteins the authors are targeting are present not only in Plasmodium, but also in other Apicomplexa parasites affecting humans and livestock. The manuscript has therefore the potential to pave way for the discovery of a new class of broad-spectrum anti-protozoan compounds which would have significant medical and veterinary implications.

Reviewer #2: This report from Russell and colleagues describes the identification and characterization of small molecule inhibitors of AP2 DNA binding factors in the human malaria parasite Plasmodium falciparum. Using an in silico screen, they identified a number of compounds that are predicted to bind the DNA interaction domain of AP2-EXP, the only AP2 for which the structure is published. The in silico hits are tested for in vitro binding of the protein and in vivo activity against cultured parasites from which, the prioritized compound is then subjected to further study to validate the mechanism of action in parasites. This study adds to the growing body of knowledge on the role of AP2 factors in Plasmodium developmental regulation, and provides preliminary validation that they may be suitable targets for antimalarial drug development.

Reviewer #3: In this manuscript titled "Identification of antimalarial compounds that inhibit Apicomplexan AP2 transcription factor proteins in the human malaria parasite Plasmodium falciparum", Russell and colleagues used in silico docking on the AP2-EXP DNA binding domain to identify potential inhibitors of ApiAP2-DNA interactions. They identified 9 compounds (A-H) based on 7 hits as potential inhibitors of ApiAP2 DNA binding that inhibited asexual parasite replication with IC50s between 11 and 170μM. Four of these (A-C,I) were able to block the interaction between AP2-EXP and DNA in vitro.In addition to AP2-EXP, the effect on DNA binding of 3 additional ApiAP2 domains were also determined. A,B,C and I inhibited DNA binding of 50%, 100%, 75% and 100% of the ApiAP2 domains tested. Two hits (A,I) were subsequently excluded because they also blocked DNA binding of either SOX2 HMGB (A) or ERF1 AP2 (I) with DNA. Four analogs of B (B1-4) were also tested for inhibition of AP2-EXP DNA binding in vitro, with B-1 and B-4 able in interfere with binding. Reported IC50 and inhibition of DNA binding did not correlate for B and its analogs. Compound C was found to arrest parasite growth at any point up to mature schizonts at 40μM. Some parasite were able to recover from exposure during the first 24h of the cycle.

Treatment with Compound C at 12μM resulted in a substantial disruption in transcription of which 10% where found to be potentially regulated by AP2-EXP based on upstream binding as determined by AP2-EXP ChIP-seq. The effect of compounds B, C and F, on oocyst maturation as also tested by injecting these into mosquitoes 10 days after blood-feeding and assaying oocyst number and oocyst spz number 4 days later. Compound C did not effect oocyst number but blocked spz formation at 18μM while compound B resulted in complete loss of oocysts at 40μM.

Unfortunately, the presented results are unable to convincingly support the assertion in the title that these compounds function by inhibiting DNA-binding in the parasite even at the micromolar concentrations used.

**Part II – Major Issues: Key Experiments Required for Acceptance**

Reviewer #1: In my opinion there are no major experiments required to complete the manuscript. There are, however, a few things that should be taken onto account while interpreting the data (see below)

Reviewer #2: Figure 3B (Lines 194-201). What was the rationale for administering all compounds at 40uM? They all have different IC50s. Is it surprising to the authors that Compound B was apparently able to completely inhibit parasite development at half the IC50 concentration reported in Figure 1?

My interpretation of the compound spike-in/wash assays is that a longer treatment time is more effective at inhibition of parasite growth. If I understand the experimental setup correctly, the treatments from Figure 3C and 3D with maximal parasite inhibition were all exposed to compound for >20 hr, compared to only 8 hours for either the 14 hpi wash out in 3C or the 46 hpi spike-in in 3D. You cannot conclude that the 30-46hpi is most sensitive to Compound C treatment without consistent time periods of exposure to the compound.

A comparison is made between the targets of AP2-EXP identified in this study by ChIP-seq and the targets of a different AP2, AP2-EXP2 (PF3D7_0611200) from Shang et al 2022. It is unclear, and not discussed by the authors how this is a meaningful comparison, particularly since their target list did not correlate with the previously published transcriptomic analysis of truncated AP2-EXP from Martins et al 2017. Based on sequence comparison or structural modeling, would AP2-EXP2 be predicted to be a target of Compounds B or C?

Reviewer #3: 1. While compounds B and C are able to block DNA binding in vitro and kill parasites, the evidence that they reduce DNA-binding in the parasite and whether this is their mechanism of action is circumstantial at best.

To the contrary, compounds B, B-1, B-2, B-3 and B-4 have very closely related structures and kill parasite with near identical IC50s (ref 38) but B-2 and B-3 do not block DNA binding/

2. It makes little sense to ascribe the effect of B & C to inhibition of AP2-EXP.

Despite the fact that compounds B and C inhibited DNA binding for 100% and 75% of the ApiAP2 domains tested

Extrapolating this observation to the 40 ApiAP2 domains across 27 proteins would suggest binding of most ApiAP2s would be impaired.

This is consistent with the observation that AP2-EXP binding was only found in 10% of down-regulated genes (Fig 19)

Also, since animals dont have AP2 domains, wouldn't a pan-ApiAP2 inhibitor be even better?

3. The stated IC50 of 80μM for B is wrong. It neither agrees with curve in S3B, not the activities of the B1-4 analogs. The graph indicates a value around 10μM, closer to the reported value in Gamo et al. (0.93μM).

The IC50 of B1-4 should be determined alongside B to allow comparison. The reported values for E (170 vs ~350), F(40 vs 70μM) also look off.

4. The phenotyping experiments (Figure 3, S13) need to be carefully redone with detailed methods and more readouts than re-invasion at a specific time-point (see comments below)

5. Figure legends largely fail to indicate the number of biological replicates, the statistical basis of the error bars, and when results are significantly different.

6. Results that are being directly compared in the text are often separated between the main and supplementary figures, making it very difficult for the reader to evaluate the support for these claims.

**Part III – Minor Issues: Editorial and Data Presentation Modifications**

Reviewer #1: 1. While all the described compounds have been designed against the AP2-EXP DNA-binding domain, it is clear that this is not their only target. Importantly there seems to be little correlation between the compound’s ability to displace AP2-EXP- bound DNA in vitro and its antiparasitic activity (IC50). The fact that the modifications that abolish the AP2 binding activity of a compound across different domains (eg B-3 variant) may even improve its lethality (fig 2) suggests that these additional targets may be proteins other than apiAP2s. While investigation of these additional targets would be very laborious and is probably out of the scope of this paper, their exitance should be acknowledged.

2. Given the above, the use of compound C to investigate the function of AP2-EXP may suffer from potential off-target effects. While I agree that the temporal correlation between the protein expression and compound activity is suggestive, it is not sufficient to claim that the resulting phenotype is the result of AP2-EXP inhibition. An interesting way to test this would be to use the compound against the asexual stages of the rodent malaria parasite. They were shown to require Ap2-exp for sporozoite development but not for asexual replication and therefore should not be affected by compound C at this stage.

3. Authors show that the compound C is active against both P.berghei and P.falciparum even if the sequences of AP2-EXP proteins and their function differ between the two parasites. Are the key residues interacting with the compound conserved between the two species? If so can the prediction be made regarding the activity of the compound against eg. P.vivax ortholog?

4. The previous work, the Western blot in fig S26 and the function of AP2-EXP, all indicate nuclear localisation. In figure S11 however the protein appears to be mostly cytoplasmic. Is that an artefact of the added tag? Or a result of protein processing? Was it observed also with the HA-tagged variant or is it unique for GFP?

5. Previous work by. Martins et al. identified Ap2-EXP as a regulator of clonally variant gene families in Plasmodium. The authors don’t mention whether this finding was confirmed using their transcriptome data. Were these genes included in the analysis?

Also – minor technical details:

6. In figure S8 3 it is unclear what are the compounds on the X-axis labelled B,K,N are (probably a leftover from an old naming system).

7. Table S3 would benefit from inclusion of the gene names or product descriptions to facilitate exploration

Reviewer #2: Figure 2. Compound B-1 has the lowest IC50 against blood stage parasites but shows poor inhibition of AP2-EXP DNA binding (compared to Compound B and B-4). While I acknowledge that there may be many explanations for differences in in vitro and in vivo activity of a molecule, the 1000-fold difference for Compound C-1 is striking. The authors should discuss this finding in a little more detail. Do they expect B-1 to bind more efficiently to another AP2 domain, or is it possible that it has other, non AP2 targets?

More detail is needed on how the growth inhibition assays were performed and IC50s calculated. The reference #63 that is provided in the Materials and Methods section does not exist. (The title is from a patent with different authors than the ones included in the reference).

Figure 3C and 3D.

The way this data is presented is not very intuitive. The authors should consider including a simple experimental overview of how compound was spiked in/washed out for each time point.

Approximately 50% of dysregulated genes after Compound C treatment were predicted targets of AP2-EXP based on the ChIP-seq assay. Are there any commonalities in the other half of dysregulated genes? In the Discussion, the authors mention that a proportion of the dysregulated genes are predicted AP-I targets but many of these are also AP2-EXP targets. Did the authors perform motif analysis on the upstream regions of the other dysregulated genes that are not targets of either AP2-I or AP2-EXP to see if there are enriched motifs that may help to predict if Compound C inhibits any other AP2 factors? Is it likely the Compound C binds multiple AP2 domains?

Reviewer #3: OTHER COMMENTS & QUESTIONS:

The introduction is completely missing background/context about previous efforts to develop small molecules that interfere with protein-DNA binding for therapeutic use in other systems.

All figures: Please ensure text in figures/table is selectable so readers can easily copy drug IDs, etc

FIG 1B:

Why is binding inhibition not quantified?

Fig 1C:

Structural similarities for A-C extend beyond benzaxolone and benzaxolone is clearly no sufficient for inhibition since D doesn't inhibit

Can you make useful inferences about SAR based on CID4541005 being inactive vs CID1365835/CID5750730 ?

Also dont all of the structures have multiple planar rings? Worth noting similarities to DNA bases?

CID for compound C is wrong, should be CID 5750730 (124220 is the TCMDC ID of the docking hit)

line 132: Is it clear that binding is competitive?

line 150: not blocking SOX2 & ERF1 doesn't demonstrate that B & C are ApiAP2 specific, merely that binding isn't broadly non-specific.

Also why disqualify I for binding ERF1 since its also an AP2 domain?

The EtBr displacement assay is not sufficient that these compounds do not bind DNA by themselves.

EtBr displacement assays were done at 50μM while EMSAs were at 300μM of the inhibitor and DNA used in EtBr assay was only 8bp long, which is less than 1 helical turn.

Moreover, intercalation is only one way that compounds can can bind DNA and block protein DNA interaction (See PMIDs: 22183179, PMID: 30774852)

The gel-shift assay from Figure 5A of PMID 16009327 or Isothermal Titration Calorimetry from https://pubs.acs.org/doi/10.1021/acs.jmedchem.8b00233 would be more suitable.

Figure S1B was a bit confusing. Switch B to D that way the analogs similar to TCMDC-124220 are A-C, and the compound similar to TCMDC123924 is D.

It would be interesting that add the predicted binding affinity

Figure S2: add structures to EC50 plots

Show binding and structure of the 2 unavailable compounds (TCMDC-124220,?

CID for compound C is wrong, should be CID 5750730 (124220 is the TCMDC ID of the docking hit)

Figure S3:

Does the >2log linear binding curve for B indicates that effect is not via DNA binding?

Combine S5 & S6.

Why do S4E&F look so different from S6A&E, even in the absence of compounds?

The structure of ERF1 AP2 bound to DNA is solved (PDB 1gcc) What are the predicted binding affinities of A-H?

Movies:

Not sure the simulations are useful, given that neither B-2 and B-3 block binding but the simulations predict that B-2 interaction is stable but B-3 is not

also Movie S2 & S5 are switched. S2 shows B-4, S5 shows B1 and Movies S3 and S5 lack the indicators for non-carbon atoms.

But if kept, switching the color is unhelpful but adding the predicted bonds formed between compound and protein might be useful visually (but not worth it if its difficult to do)

What was the point of B1-4 if follow-up is done with B?

Figure 3 / Figure S13 / Figure S14

Figure S13, S14 results should be included in Fig 3.

If the goal of this figure is to show that B & C have similar effects on parasites then the same phenotyping experiments need to be done for B as were done for C in S14 and 3C-D.

Figure S13: I could not find the starting parasitemia for this experiment in the methods, but based on S13A, it would have been set up at 8% trophs if parasite are schizonts at 24h post spike-in.

That seems like very high starting parasitemia, when growth is to be maintained for another 72h.

The parasitemia of the DMSO treated fails to increase ( 8% at 24h post spike-in, 7% at 48h post spike-in, 9% at 72h post spike-in)

The figure shows parasitemia at 24, 48, 72h after treatment not just 48 as indicated in the figure title.

The cell shown in Fig3B for H-treated cultures at 48h-post spike-in is a gametocyte (note the dispersed hemozoin), is this representative?

Figure S13/14: Phenotypes dont agree. Figure 13 shows B & C-treated parasite are rapidly lost, while in S14 parasitemia is constant in C-treated cultures.

Figure 3C-D, why is the ring parasitemia in the next cycle the only read-out from this experiment? Are parasites killed or arrested?

Line 191: Just because compounds arrest at at similar point doens't mean they are likely to have the same target.

The fact that the inhibition of DNA binding and IC50s dont seem to be correlated (B, B1-4 have very similar IC50s between 0.25-1.1μM), would suggest that inhibition of AP2-EXP binding isn't the mechanism.

Line 197: states that parasites fail to progress but based on S13, they are actually disappearing from the culture

Lines 575-589: unclear, write separate descriptions of the spike-in and wash-out experiments.

Based on this description, adding drug and immediately washing it out (14hpi timepoint), reduces parasitemia by 50% in the next cycle.

Is that correct? If so, it seems surprising that exposure for 8h (22hpi washout) doens't have much more profound effects, than exposure for only a few minutes.

Are the methods possibly incorrect and C was added at 6hpi (as for S14) and the 14hpi represents the effect of 8h drug exposure?

Figure 3C: How do the authors explain that AP2-EXP is not expressed at 18hpi but exposure to C until 14hpi and 22hpi already has substantial impact parasite growth?

Figure 3D: What was the ring-stage parasitemia in cultures when compound C was added at 46hpi? A substantial fraction of the rings at 54hpi might already have been present at 46hpi.

FIGURE 4

Figure 4A, S15D: The correlation matrices between DMSO-treated time-points (DMSO on both x & y axis) needs to also be shown in order for these figure to be interpretable since it is the difference between these correlation matrices that indicates the change.

Minor point: DMSO / compound C-treated are shown on opposite axes in 4A and S15D. It took me a second to realize that the matrix wont change. To avoid this confusion in other readers, it would be best to keep them on the same axis.

Lines 224-229:

Correlation between DMSO and C-treated timepoints is lower when all transcripts are used than when only ~50 highly periodic genes are used.

The text states that this indicates that there isn't an overall perturbation of the transcriptome.

However, this would make sense if only a small subset of genes exhibited altered expression but S19 shows that at the 24-30h time-point alone, expression of nearly 10% of parasite genes was altered by at least 2x.

Since, 80% of genes are developmentally regulated and the the number of genes with altered expression greatly exceeds the number genes in the control set, it seems that the opposite is the case: the overall transcriptome is more perturbed than highly periodic genes.

Figure 4B: Is the x-axis supposed to show the motif distance to the start codon?

Figure 4C: The grey color is supposed to indicated "Non Significant" but genes in green are also non-significant.

Also, based on the Table S3, the y-axis shows the -log10(Adjusted P-value), not the -log10(P-value).

Figure S18: Legend states that the Control (-) track shows the coverage but since coverage can't be negative this must also be enrichment when IP-ing from an untagged line.

Which antibody was used, anti-GFP or anti-HA?

Full ChIP-seq (bigwig or bedgraph) tracks need to me made available.

Figure 5: How did the size and morphology of C-treated oocyst differ from DMSO-treated oocysts.

PLOS authors have the option to publish the peer review history of their article (what does this mean?). If published, this will include your full peer review and any attached files.

Reviewer #1: No

Reviewer #2: No

Reviewer #3: No
---

## [Decision Letter · Decision Letter 1]

17 Sep 2022

Dear Dr. Russell,

We are pleased to inform you that your manuscript 'Inhibitors of ApiAP2 Protein DNA Binding Exhibit Multistage Activity Against Plasmodium Parasites' has been provisionally accepted for publication in PLOS Pathogens.

Best regards,

Laura J. Knoll

Pearls Editor and Guest Associate Editor

PLOS Pathogens

Kami Kim

Section Editor

PLOS Pathogens

Kasturi Haldar

Editor-in-Chief

PLOS Pathogens

orcid.org/0000-0001-5065-158X

Michael Malim

Editor-in-Chief

PLOS Pathogens

orcid.org/0000-0002-7699-2064

Reviewer Comments (if any, and for reference):

Reviewer's Responses to Questions

**Part I - Summary**

Reviewer #1: All my concerns have been addressed adequately. I have no further comments.

Reviewer #2: The authors provide a much improved, revised manuscript. More detailed methods and statistical analyses have been included. The calculated IC50 values for the inhibition assays have been corrected, based on the original inhibition data that was presented in the original submission. New, more thorough phenotyping experiments have been performed to determine the point in the replicative cycle that is most sensitive to compound treatment. This experiment was also modified for consistent exposure times of the parasites to the compound at each time point tested. Supplementary materials also include a helpful experimental outline which makes interpretation of the data much more intuitive.

Importantly, the discussion acknowledges that the experimental inhibitors tested here likely target multiple AP2 factors and may even have additional off-target effects. This is a more appropriate conclusion based on the data they have collected and present here.

Reviewer #3: The revised manuscript is improved. And while it clearly shows that molecular docking can identify molecules that can block ApiAP2 interactions in vitro, the evidence linking the activity of these compounds to disrupting those interactions in the parasite remains correlative at best and at times not even that. The compound B demonstrate this disconnect between their ability to block ApiAP2 binding and anti-parasite activity. Compounds B (EC50 18μM), B-1 (0.16μM) and B-4 (1μM) can all block DNA binding differ 100-fold in activity, while B-2 (1μM) and B-3 (1μM) are unable to block binding but have EC50s squarely in the middle of the range of those that can block binding.

**Part II – Major Issues: Key Experiments Required for Acceptance**

Reviewer #1: (No Response)

Reviewer #2: NA

Reviewer #3: (No Response)

**Part III – Minor Issues: Editorial and Data Presentation Modifications**

Reviewer #1: (No Response)

Reviewer #2: NA

Reviewer #3: (No Response)

PLOS authors have the option to publish the peer review history of their article (what does this mean?). If published, this will include your full peer review and any attached files.

Reviewer #1: No

Reviewer #2: No

Reviewer #3: No

---

## [Editor Report · Acceptance letter]

3 Oct 2022

Dear Mr. Russell,

We are delighted to inform you that your manuscript, "Inhibitors of ApiAP2 Protein DNA Binding Exhibit Multistage Activity Against Plasmodium Parasites," has been formally accepted for publication in PLOS Pathogens.

Best regards,

Kasturi Haldar

Editor-in-Chief

PLOS Pathogens

orcid.org/0000-0001-5065-158X

Michael Malim

Editor-in-Chief

PLOS Pathogens

orcid.org/0000-0002-7699-2064